# A fine-scale *Arabidopsis* chromatin landscape reveals chromatin conformation-associated transcriptional dynamics

Yueying Zhang [1,2,6], Qianli Dong [1,6], Zhen Wang [2,3,6], Qinzhe Liu [4,5,6], Haopeng Yu [2], Wenqing Sun[1], Jitender Cheema[2], Qiancheng You[4,5], Ling Ding[1], Xiaofeng Cao [3], Chuan He [4,5], Yiliang Ding [2] ✉ & Huakun Zhang [1] ✉

Plants, as sessile organisms, deploy transcriptional dynamics for adapting to extreme growth conditions such as cold stress. Emerging evidence suggests that chromatin architecture contributes to transcriptional regulation. However, the relationship between chromatin architectural dynamics and transcriptional reprogramming in response to cold stress remains unclear. Here, we apply a chemical-crosslinking assisted proximity capture (CAP-C) method to elucidate the fine-scale chromatin landscape, revealing chromatin interactions within gene bodies closely associated with RNA polymerase II (Pol II) densities across initiation, pausing, and termination sites. We observe dynamic changes in chromatin interactions alongside Pol II activity alterations during cold stress, suggesting local chromatin dynamics may regulate Pol II activity. Notably, cold stress does not affect large-scale chromatin conformations. We further identify a comprehensive promoter-promoter interaction (PPI) network across the genome, potentially facilitating co-regulation of gene expression in response to cold stress. Our study deepens the understanding of chromatin conformation-associated gene regulation in plant response to cold.

As sessile organisms, plants have developed adaptive strategies to thrive under diverse environmental pressures, including harsh growth conditions. In recent years, significant effort has focussed on unraveling the molecular mechanisms that underlie plant response and adaptation to environmental stress[1]. Cold stress is a critical factor that affects plant growth and development, often limiting the geographical range of plant species. During cold stress, plants undergo global transcriptional reprogramming, resulting in the upregulation or downregulation of many genes[1]. In addition to changes in steady-state gene expression levels, a recent study has shown that Pol II activities during transcription elongation, such as Pol II pausing, are dynamically altered in response to cold[2]. It has been suggested that Pol II activities,

including Pol II pausing, are closely linked to chromatin organization[3]. However, it is poorly understood whether changes in chromatin dynamics occur in response to cold and contribute to this global transcriptional reprogramming.

Furthermore, among all upregulated or downregulated genes, genes associated with metabolic pathways and photosynthesis were upregulated or downregulated in parallel, alongside those genes involved in cold response[1]. These genes are suggested to be co-regulated by cold-responsive transcription factors in balancing the trade-off between plant growth and cold tolerance[4]. However, it is unclear how these transcription factors can co-regulate different genes with distinct functions. Chromatin interactions may offer a potential

[1]Key Laboratory of Molecular Epigenetics of Ministry of Education, Northeast Normal University, Changchun 130024, China. [2]Department of Cell and Developmental Biology, John Innes Centre, Norwich Research Park, Norwich NR4 7UH, UK. [3]Institute of Genetics and Developmental Biology, Chinese Academy of Sciences, 100101 Beijing, China. [4]Howard Hughes Medical Institute, The University of Chicago, Chicago, IL 60637, USA. [5]Department of Chemistry, Department of Biochemistry and Molecular Biology, Institute for Biophysical Dynamics, The University of Chicago, Chicago, IL 60637, USA. [6]These authors contributed equally: Yueying Zhang, Qianli Dong, Zhen Wang, Qinzhe Liu. ✉e-mail: yiliang.ding@jic.ac.uk; zhanghk045@nenu.edu.cn

explanation for how transcription factors can co-regulate genes with distinct functions by bringing together different regions of the genome.

The latest advances in high-throughput chromatin conformation capturing technologies, such as Hi-C, have revolutionized our comprehension of chromatin architecture in plants[5,6]. However, Hi-C with its several kilobase (kb) resolution along with protein-mediated crosslinking has limited capability for exploring either chromatin conformations at the gene level or promoter-promoter interaction (PPI) networks at the genome-wide scale. To decipher whether chromatin interactions contribute to transcriptional reprogramming in plant response to cold, here, we leveraged chemical-crosslinking assisted proximity capture (CAP-C) methodology that directly cross-links DNAs[7]. CAP-C provided a chromatin contact map with high resolution and low background noise[7]. Thus, we generated the first fine landscape of *Arabidopsis* chromatin interactions with high resolution and accuracy. Our CAP-C determined that chromatin interactions were significantly associated with Pol II activities at the gene level. We observed chromatin dynamics coupled with transcriptional reprogramming in response to cold treatments (3-h and 12-h at 4 °C). We also captured a comprehensive PPI network primed in the genome that may facilitate gene expression co-regulation in response to cold. Together, our study suggested that plants adopted distinct and multilevel chromatin conformations in tuning transcriptions in response to cold.

## Results

### CAP-C successfully captured global chromatin conformational features at high resolution and low background noise in *Arabidopsis*

To examine high-resolution chromatin conformational features at the gene level, we applied chemical-crosslinking assisted proximity capture (CAP-C). This method enables the identification of short-range interactions without the mediation of proteins using a new type of crosslinker, poly (amidoamine) (PAMAM) dendrimers[7]. A small-sized dendrimer favors finer interactions[7]. Here, we used the smallest G3 dendrimer (3.6 nm diameter) to obtain a fine chromatin conformational map in *Arabidopsis* (Fig. 1a). Nuclei from 10-day-old *Arabidopsis* seedlings growing at 22 °C were extracted and incubated with dendrimers. After UV crosslinking the dendrimers with double-stranded DNAs (Fig. 1a and Supplementary Fig. 1a), the complex was digested into ~200 bp. Bifunctional bridging linkers were added to the reaction, ligating to dendrimers via "click chemistry"[8]. The ligated products were enriched by capturing the biotin on the bridge linker before subjecting to library construction and high-throughput sequencing (Fig. 1a). We termed the read pairs that contain the bridge linker in the middle as the valid "chromatin contacts". Two independent biological replicates were generated with high reproducibility (Supplementary Fig. 1b).

To verify the efficacy of our CAP-C approach, we first evaluated the overall characteristics of chromatin conformations. Prior cytological investigations revealed that the chromatin architecture of *Arabidopsis* exhibits significant interaction patterns that encompass pericentromeres and telomeres[9–11]. Our CAP-C data showed strong chromatin contacts at both telomeric ends of all five chromosomes (Fig. 1b, upper panel), consistent with previous Hi-C data[11,12]. Notably, our CAP-C also detected strong chromatin contacts in centromeric regions (Fig. 1b, lower panel) that were not detectable in Hi-C[11,12]. Further comparisons across the centromeric regions between CAP-C and Hi-C at different resolutions (20 kb, 10 kb, and 2 kb) (Fig. 1c) revealed that our CAP-C chromatin contacts signals were stronger and finer with lower background noise compared to Hi-C (Fig. 1c). Our CAP-C clearly captured chromatin contacts toward ~200 bp resolution, enabling us to identify local chromatin contacts (Supplementary Fig. 1c). We further assessed the effectiveness of our CAP-C by averaging the genome-wide contact frequency. In contrast to Hi-C data[12] (Supplementary

Data 2), our CAP-C data showed more short-range chromatin contacts (Fig. 1d). Together, our results confirmed that we had successfully applied CAP-C in *Arabidopsis*, generating a fine chromatin conformational map at ~200 bp resolution with low background noise.

### CAP-C efficiently identified local chromatin conformational features in *Arabidopsis*

Our high-resolution chromatin contact map enabled us to characterize local chromatin organizational features and to validate these features on individual gene loci, enhancer-promoter (E-P) and promoter-promoter (P-P) interactions. To facilitate comparison with the previous studies[13,14], we used Fit-Hi-C[15] to identify highly enriched chromatin contacts (termed as stable chromatin contacts). Previously reported chromatin interactions across the *FLOWERING LOCUS C* (*FLC*) locus[12,14] were identified from our CAP-C (Fig. 2a, red curve) data. Additionally, we identified short-range stable chromatin contacts enriched in the first intron region of *FLC* proposed to be associated with promoter-proximal Pol II pausing[16]. Another interaction loop between the promoter region of *PINOID* (*PID*) and the loci of lnc RNA *APOLO* (*AT2G34655*)[13,17] was also apparent in our CAP-C (Fig. 2b, red curve) data, further supporting that our CAP-C accurately captured chromatin contacts at the gene level.

E-P interactions are another form of fine-scale chromatin contacts[18]. As critical regulatory components, enhancers can regulate gene expression through E-P interactions[19–21]. To verify the accuracy of our CAP-C data, we took advantage of the global nuclear run-on sequencing (GRO-seq) data that defined active enhancer sites across *Arabidopsis* genomes[22]. We observed that chromatin contacts were highly enriched across active enhancer sites (Supplementary Fig. 2a). It was suggested that the *AT3G03850* (*Small auxin up RNA26*, *SAUR26*) forms E-P interactions with those enhancers with the E-box (CANNTG)-like motif CAAGT(T/G)G, targeted by a transcription factor KANADI1 (*KAN1*)[23]. Here, we identified that one active enhancer with the E-box (CANNTG)-like motif CAAGT(T/G)G exists immediately downstream of the *SAUR26*. We found this active enhancer site that was highly transcribed and associated with active histone markers such as H3K36me3, H3K4me3, and H3K9ac (Fig. 2c). This active enhancer locus directly interacted with the promoter region of the *SAUR26*, leading to its associated high expression (Fig. 2c). Another example of the chromatin interaction between an active enhancer site and the promoter region of its adjacent gene *AT1G31910* (*GHMP kinase*) is illustrated in Fig. 2d. Therefore, these results further support that our CAP-C is robust and sensitive. We further conducted comprehensive characterizations of E-P chromatin contacts (Supplementary Data 8 and 9). Previous studies showed that enhancers work as transcriptional regulatory elements, orchestrating gene expression from a distance[24,25]. We then conducted an analysis on the distribution of the distances for all E-P chromatin contacts determined in our CAP-C libraries in all the chromosomes (Supplementary Fig. 3 and Supplementary Data 8 and 9). We found that the majority (89%) of E-P chromatin contacts were distal while 11% were proximal, aligning with previous findings[24,25]. Among proximal contacts, 62% were upstream and 38% downstream of the interacting promoter, while distal contacts exhibited 51% upstream and 49% downstream orientation (Supplementary Fig. 4 and Supplementary Data 8). Taken together, our CAP-C detected both proximal and distal E-P chromatin contacts with a predominant tandem orientation.

Our CAP-C was able to capture interactions between different gene loci. Several genes engaging in many intra-chromosomal interactions using Bridge Linker-Hi-C (BL-Hi-C) were reported as a single gene locus interacting with multiple loci across the same chromosome[26]. Intra-chromosomal interactions representative of protein-coding genes *AT1G58602* (*Recognition of Peronospora Parasitica 7*, *RPP7*) and *AT1G01320* (*Reduced Chloroplast Coverage 1*, *REC1*) are illustrated in Fig. 2e, f, respectively. *AT3G56825* and *AT4G13495*,

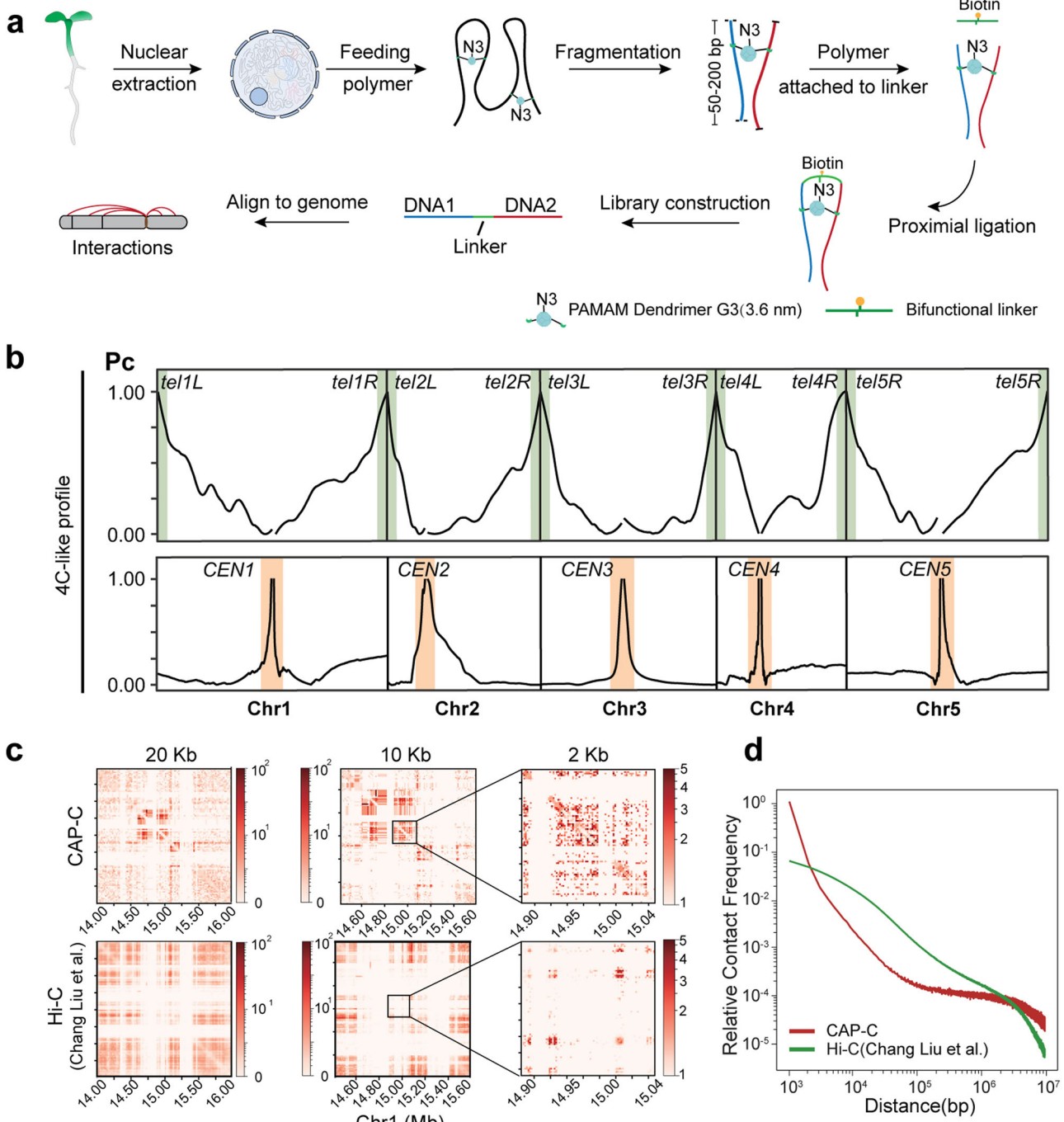

**Fig. 1 | CAP-C for the high-resolution chromatin conformations in *Arabidopsis*.**
**a** Shown schematically here, the CAP experimental procedure involved extracting nuclei from seedlings and incubating them with dendrimer. Multi-functional dendrimers and chromatin were photo-crosslinked by UV light irradiation. Following global fragmentation, the azide handle (N3 group) attached to the dendrimer-chromatin complex was then reacted with a bi-functional bridge linker oligonucleotide for two-step proximity ligation. The resulting chimeric fragments (one end labeled blue, another end labeled red, with the biotin-labeled bridge linker in the middle) were then purified for library construction, sequencing, and data analysis to obtain chromatin contact profiles. **b** 4C-like inter-chromosomal interaction profiles showing the average contact probabilities across the centromeres and telomeres regions for each chromosome. **c** Normalized contact matrix maps of CAP-C (upper panel) and Hi-C[12] (bottom panel) at various resolutions of the centromeric region of chromosome 1. **d** Relative contact frequency versus genomic distance curves of CAP-C and Hi-C[12]. *X*-axis: the distance between contact loci from 1 kb to 10 Mb. *Y*-axis: contact density normalized by sequencing depth.

which are responsible for transcribing snRNA *U2.4* and lncRNA, respectively, are two additional gene examples that contain numerous known intra-chromosomal interactions (Supplementary Fig. 2b, c). Our CAP-C not only verified the interactions previously reported by BL-Hi-C but also enabled us to identify more short-range interactions. We closely examined these interactions and identified another fine-scale type of chromatin contact known as promoter-promoter interaction

(PPI). PPI pinpointed the promoter region of a particular gene interacting with the promoter region(s) of one or more other genes[27]. With regards to *RPP7* and *REC1*, they interacted with 108 and 120 genes, respectively, on the same chromosome that engaged in PPIs (Fig. 2e, f). These PPIs may facilitate the RNA-mediated chromatin interactions reported at these loci[26]. Thus, our CAP-C data has successfully shown genes engaging in many distinguishable intra-chromosomal

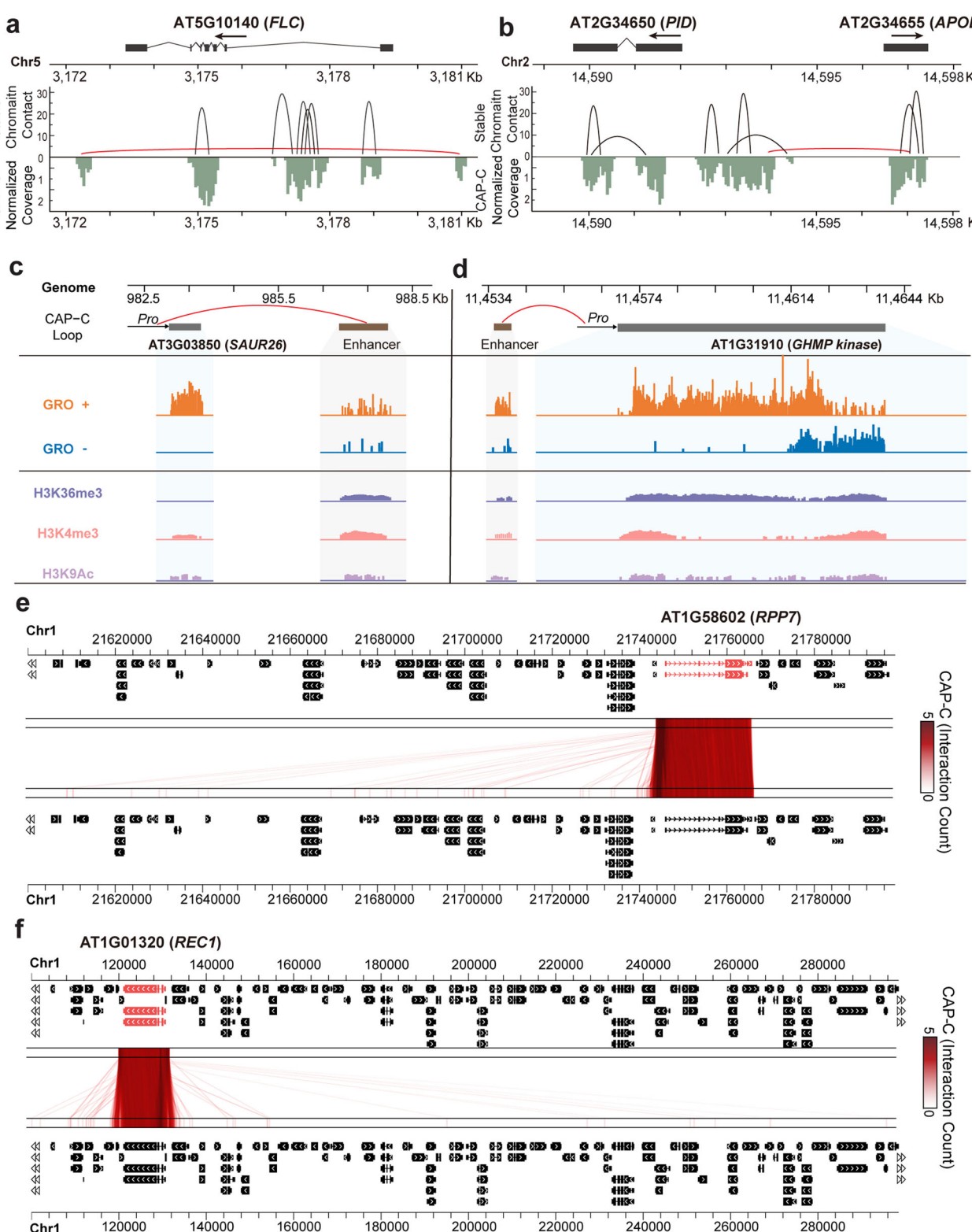

interactions, supporting the high resolution and sensitivity of our CAP-C method. Our collective findings suggested that our CAP-C approach can accurately and sensitively detect intricate chromatin conformational characteristics.

## Local chromatin contacts at the gene level are significantly associated with Pol II activities

Local interactions at the gene level are proposed to be closely linked with RNA polymerase II (Pol II) activities[18,28]. In our CAP-C study, we

examined the association between chromatin contacts and Pol II activities. As nascent RNA expression levels can indicate overall Pol II activities[22], we investigated our CAP-C chromatin data and found that gene chromatin contacts are significantly associated with transcriptional levels (Fig. 3a, Spearman's $R_s$ = 0.39). We have also observed an association between chromatin accessibilities and local chromatin interactions as measured by our CAP-C (Supplementary Fig. 5 and Supplementary Data 2). We further explored relationship between representative histone modifications/variants and chromatin contact

**Fig. 2 | Characterization of chromatin conformations. a, b** The stable chromatin contacts around *FLC* (**a**) and *PID* and *APOLO* (**b**) loci. The schematic gene structure was aligned to stable chromatin contacts (loops) captured by CAP-C and coverage results, with boxes representing exons and lines representing introns. The red curve shows the reported chromatin contacts validated experimentally, and the black curves show the newly identified stable chromatin contacts by CAP-C, associated with which are the CAP-C reads shown under the chromatin contacts.
**c, d** Representative examples demonstrating enhancer-promoter interactions. Black boxes represent gene loci, arrows show the promoter regions and transcription directions. Brown boxes represent active enhancer loci. Interactions

between an enhancer and a promoter are shown in the red curve. GRO-seq reads (GRO+ represents the sense strand, while GRO− represents the antisense strand) and the corresponding histone ChIP-seq tracks (H3K36me3, H3K4me3, H3K9Ac) were shown underneath each enhancer and gene locus. GIVE plots showing representative short-range (**e**, *RPP7*) and long-range (**f**, *REC*) intra-chromosomal DNA-DNA interactions based on our CAP-C data under normal growth conditions. To depict interactions, the chromosomes were plotted horizontally twice, producing a top and a bottom track. Each red line represents a chromatin contact, with the color scale indicating the strength of the contacts. The black boxes and lines indicate genes on the chromosomes.

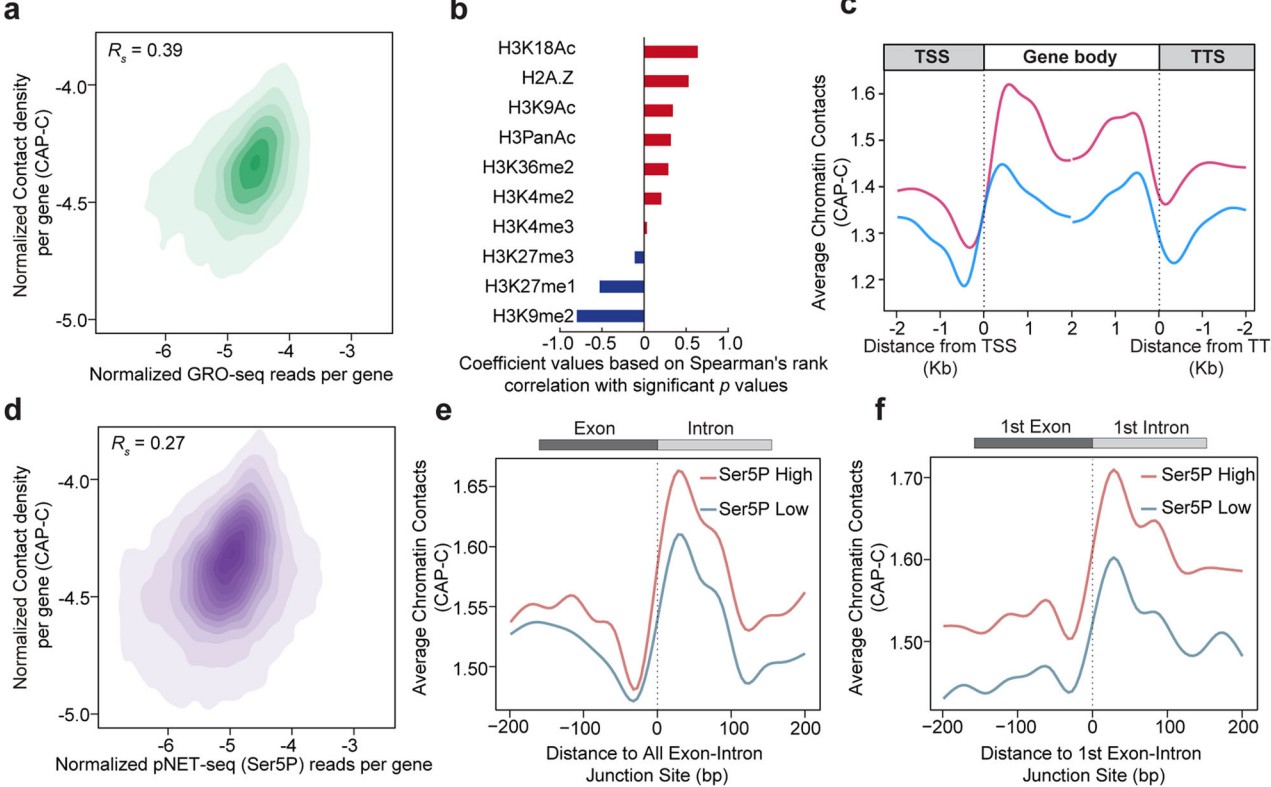

**Fig. 3 | CAP-C association between local chromatin contacts and Pol II activities. a** Correlation between gene chromatin contacts with transcriptional activities. The gene chromatin contact was measured by normalized chromatin contact density per gene. The transcription activity was measured by GRO-seq[22] signal enrichment. The correlation between gene chromatin contacts and Pol II activities was 0.39, *p* value = 8.150194e$^{-224}$, two-sided Spearman's rank correlation. **b** Based on Spearman's rank correlation, coefficient correlations with significant *p* values between CAP-C gene chromatin contacts and different histone modifications are shown, two-sided Spearman's rank correlation. **c** Meta-profile shows CAP-C interaction signal profiles of highly (pink) and weakly (light blue) transcribed genes, as shown by 10% of the highest (*n* = 2739) and lowest (*n* = 2769) transcribed genes,

which are calculated using GRO-seq[22]. mRNAs were aligned by the transcription start site (TSS) and the transcription termination site (TTS). **d** Correlation between gene chromatin contact and Pol II elongation activity. The gene chromatin contact was measured by normalized chromatin contact density per gene. The Pol II elongation activity was measured by Ser 5p Pol II pNET-seq[22]. The correlation between gene chromatin contacts and Ser 5p Pol II activities was 0.27, *p* value < 1.7e$^{-308}$, two-sided Spearman's rank correlation. Meta-profiles show the CAP-C chromatin contacts signal profile of 10% of the highly (red, *n* = 10,663) and weakly (blue, *n* = 10,511) Ser 5p Pol II activities across the (**e**) 5′SS or (**f**) the 1st 5′SS, as shown by 10% of the highest (*n* = 1933) and lowest (*n* = 1944) Ser 5p Pol II activities.

density at the gene level. Our results demonstrate that active histone modifications, including H3K18Ac, H2A.Z, H3K9Ac, H3PanAc, H3K36me2, H3K4me3, and H3K4me2, are significantly linked with gene chromatin contacts (Fig. 3b and Supplementary Data 2 and 3). Moreover, repressive modifications, such as H3K27me3, H3K27me1, and H3K9me2, are negatively associated with gene chromatin contacts (Fig. 3b and Supplementary Data 2 and 3).

Notably, histone modifications were shown to affect Pol II elongation during transcription[29]. Thus, we examined the distribution of these chromatin contacts across the gene bodies. We ranked our mRNAs according to expression levels and found that highly expressed genes contain overall more chromatin contacts across the gene bodies compared to weakly expressed genes (Fig. 3c). In particular, there were

more chromatin contacts around transcription start sites (TSSs) and transcription termination sites (TTSs) (Fig. 3c) where transcriptional activities were higher[22]. We then explored the regions between TSS and TTS by dividing genes into groups with and without introns. We found that chromatin contacts were significantly more intense in those genes containing introns than those without (Supplementary Fig. 6a). After partitioning the exons and introns, we observed that the frequency of chromatin contacts within introns is significantly greater than those within exons (Supplementary Fig. 6b). Notably, the chromatin contacts were not evenly distributed across exon and intron regions but rather were enhanced at the start and end of the introns (Supplementary Fig. 6c). This feature reminds us of Pol II pausing signatures across the 5′ Splicing Site (5′SS) located ~25 bp downstream of the 5′SS[2]. To

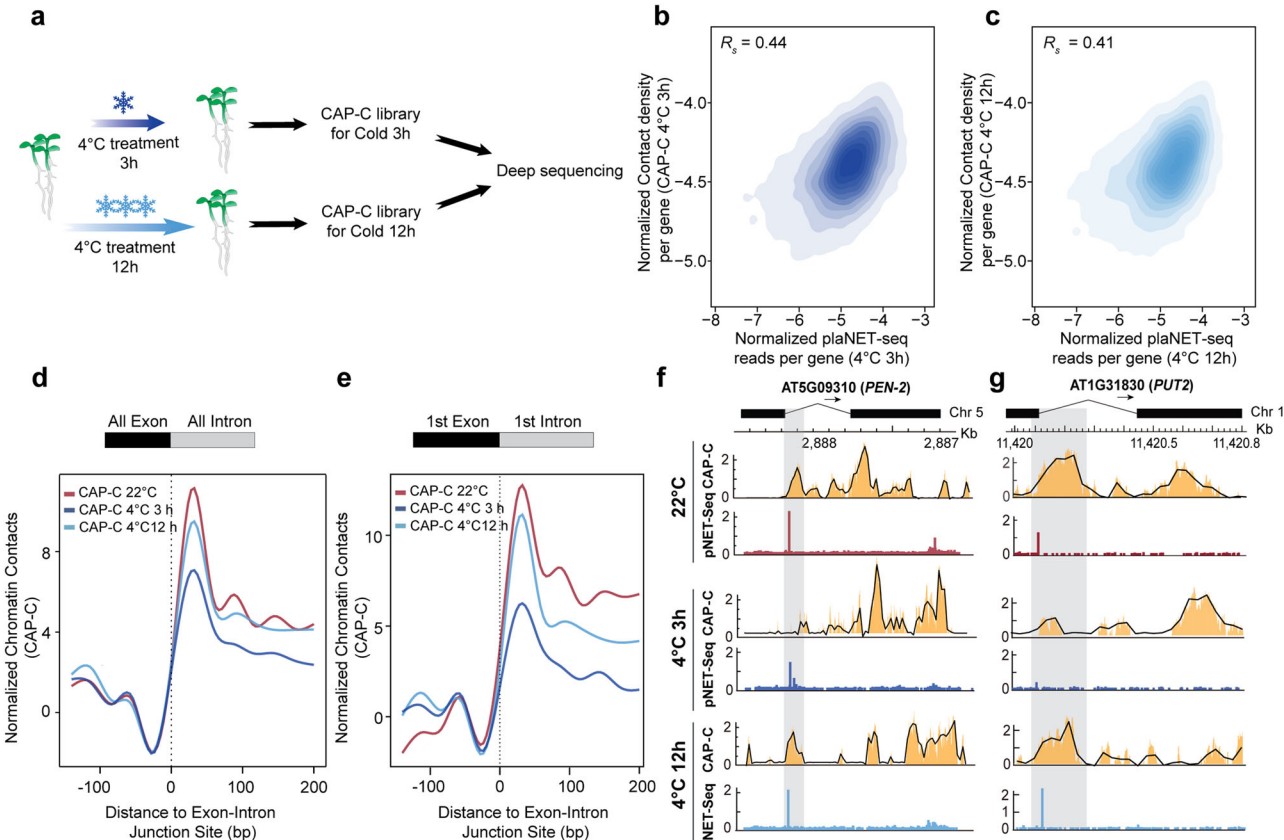

**Fig. 4 | Chromatin contacts dynamically coordinate with Pol II activities in response to cold stress. a** Schematic illustration of plant cold treatments. 10 day-old seedlings were exposed to low temperature (4 °C) for 3 or 12 h. Samples were collected for CAP-C library constructions. **b, c** Correlations between gene chromatin contacts and transcriptional activities under 3 h and 12 h cold treatments, respectively. The gene chromatin contact was measured by normalized chromatin contact density per gene. The transcription activity was measured by the plaNET-seq[2]. The correlation between gene chromatin contacts and Pol II activities was

0.44, *p* value < 1.7e−308 for 3 h cold treatment and 0.41, *p* value < 1.7e−308 for 12 h cold treatment, two-sided Spearman's rank correlation. Meta-profiles show the CAP-C chromatin contact profile 100 bp upstream and 200 bp downstream of the (**d**) 5'SS or (**e**) the 1st 5'SS (22 °C in red, 4 °C treatment for 3 h in dark blue, 4 °C treatment for 12 h in light blue). Z-score normalization was applied to normalize the difference of the sequencing depths in the libraries of three conditions. Examples show the chromatin contacts and Pol II density signal profiles across the gene AT5G09310 (*PEN-2*) (**f**) and AT1G31830 (*PUT2*) (**g**).

investigate any association between gene-body chromatin contacts and Pol II pausing signatures, we measured the correlation between CAP-C gene chromatin contacts with Pol II activities across the gene body. The phosphorylation of Ser5 inside the carboxyl-terminal domain (CTD) representing transcriptional activity during Pol II elongation is well-documented[28]. Notably, we found a significant positive correlation (Fig. 3d, Spearman's $R_s$ = 0.27) between gene chromatin contacts and Ser5P Pol II density, indicating a potential link between these two features. After ranking Ser5P Pol II density, we investigated the chromatin contact frequency across exon-intron junction sites. Our results indicated that the regions with high Ser5P Pol II densities exhibited more chromatin contacts than those with low Ser5P Pol II densities (Fig. 3e). Notably, this disparity was most prominent at the first exon-intron junction, which is critical to Pol II elongation activity (Fig. 3f). Our findings thus shed light on pervasive associations between chromatin contacts and Pol II activities during initiation, elongation, and termination at the gene level.

**Chromatin conformation-associated Pol II activities are dynamically altered in plant response to cold**

A previous study found that the Pol II densities across exon-intron junction sites dynamically changed in response to cold treatments (3-h and 12-h at 4 °C)[2]. The Pol II pausing signal dropped rapidly during cold treatment for 3 h and gradually restored during cold treatment for 12 h[2]. This result suggested a dynamic

reprogramming of nascent Pol II activities in plant response to cold stress. Thus, we speculated whether these transcriptional dynamics of Pol II activities during elongation were linked to chromatin conformations. To test this, we generated CAP-C libraries corresponding to the same cold treatments of seedlings at 4 °C for 3 and 12 h, respectively (Fig. 4a, Supplementary Fig. 1b and Supplementary Data 1). We compared the correlation between gene chromatin contacts obtained from CAP-C libraries and Pol II densities derived from previous plant native elongating transcripts sequencing (plaNET-seq) libraries for varying cold treatment durations (Fig. 4b, c, Supplementary Fig. 7a and Supplementary Data 2). We found significant positive correlations between gene chromatin contacts and Pol II densities at normal temperature (22 °C), and at 4 °C for 3 h and 4 °C for 12 h, respectively (Fig. 4b, c and Supplementary Fig. 7a Spearman's $R_s$ = 0.47, Spearman's $R_s$ = 0.44, Spearman's $R_s$ = 0.41, respectively). These findings suggested that the relationship between chromatin contacts and Pol II activities at the gene level was likely to be pervasive across diverse conditions.

Next, we assessed chromatin contact densities across exon-intron junction sites and found that the changes in chromatin contacts were notably similar to Pol II density changes under cold treatments (Fig. 4d, e and Supplementary Fig. 7b, c). Specifically, the chromatin contacts located downstream of 5'SS dropped sharply during 3 h in the cold, but gradually reverted to control levels during 12 h in the cold (Fig. 4d). In

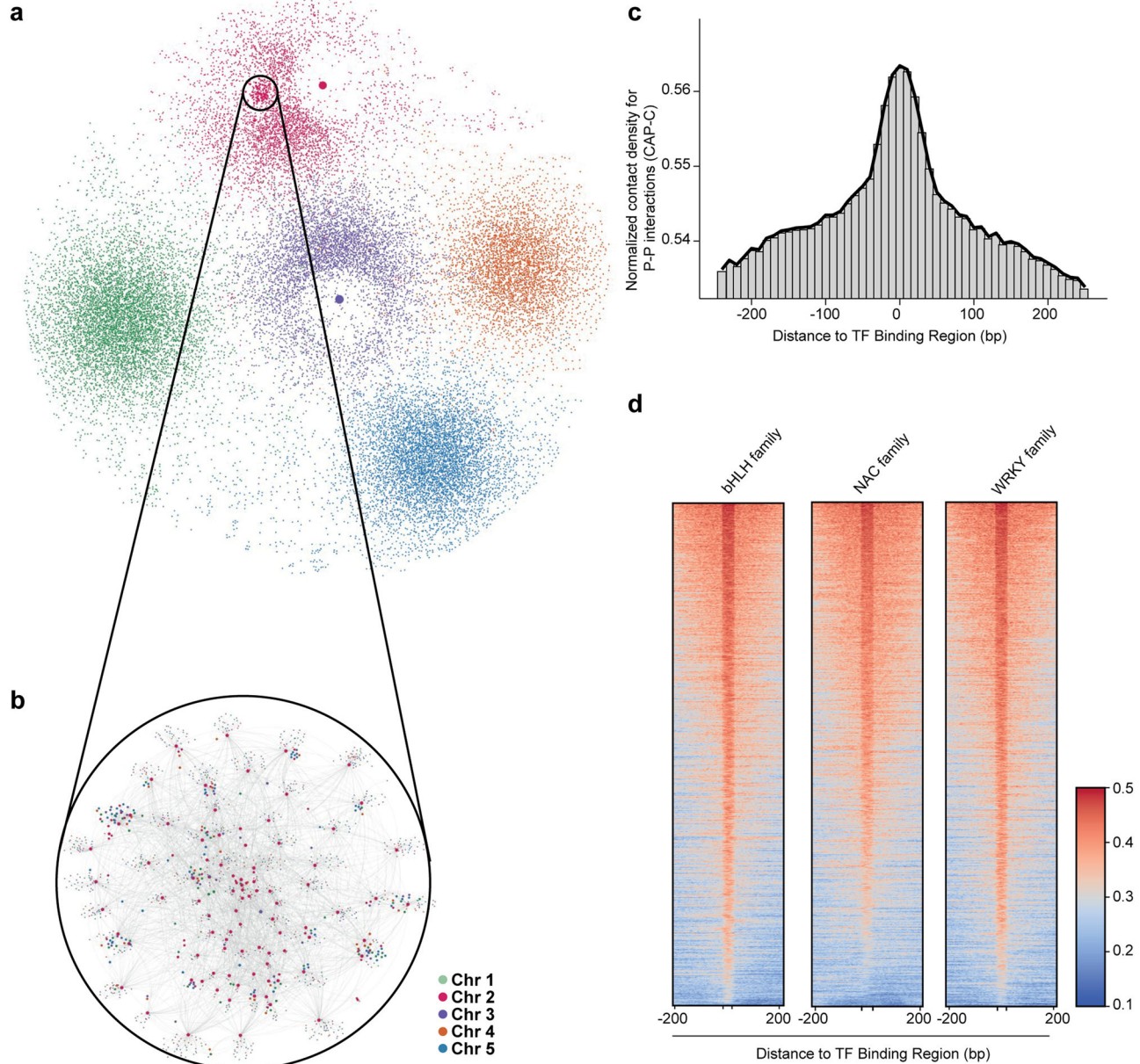

**Fig. 5 | The primed promoter-promoter interaction network may harbor transcription factors. a** A genome-wide promoter-promoter interaction (PPI) map of *Arabidopsis* for the PPIs present under three temperature conditions was displayed using Cytoscape 3.9.1[50]. The dark pink and purple dots inside the network were two PPI hotspots in chromosomes 2 and 3, respectively. The five individual chromosomes were shown in different colors. The size of the dots indicated the number of PPIs. **b** An enlarged region of the PPI network on chromosome 2. **c** Histogram of chromatin contacts over the PPIs across the binding sites of 422 transcription factor (TF). TF binding sites were obtained from previous ChIP experiments[31,52] (see Supplementary Data 2). **d** Heatmap showing strengths of PPIs across the binding sites of the bHLH, NAC, and WRKY transcription factor families, respectively.

particular, these dynamic changes of local chromatin contacts were more prominent across the first exon-intron junctions (Fig. 4e). This associative evidence between local chromatin interactions and Pol II activities implied that local chromatin conformations may facilitate transcriptional reprogramming in plant response to cold stress. Example associations across the first 5′SS of genes *AT5G09310* (*Probable gamma-secretase subunit, PEN-2*) and *AT1G31830* (*Paraquat resistant 2, PUT2*), between corresponding dynamic changes of both local chromatin contacts and Pol II activities, are illustrated in Fig. 4f, g. Hence, our findings suggested that during extreme growth conditions, such as cold stress, dynamic changes in local chromatin contacts within the gene locus occur that may contribute to the reprogramming of transcriptional activities.

## CAP-C revealed a comprehensive PPI network that pervaded the genome

After discovering alterations of local chromatin conformations in response to cold, we then assessed large-scale chromatin conformations. We found that most large-scale chromatin conformations remain stable (Supplementary Fig. 8). We then investigated PPIs across the whole genome and identified PPIs that existed concurrently in three sets of CAP-C libraries under different temperature conditions that were likely to be thermo-stable and primed. Notably, we found a considerable number of PPIs, including intra-chromosomal and inter-chromosomal interactions, across all five chromosomes, forming a comprehensive PPI network (Fig. 5a, b). Furthermore, two PPI network hotspots were determined (Fig. 5a) connecting one gene locus to

multiple genes. One of these hotspots was found on chromosome 2, where the promoter of *AT2G01023* interacted with the promoters of 4827 genes. The other hotspot was situated on chromosome 3, where the promoter of *AT3G41768* (rDNA) interacted with the promoters of 5407 genes (Fig. 5a). These two PPI networks might be primed to assist the extensive RNA-mediated chromatin contacts in these loci[26]. We also did a comprehensive analysis on the PPIs (Supplementary Fig. 9 and Supplementary Data 10 and 11). A notable majority (77%) of these interactions occurred in a *cis* manner, where interactions take place on the same chromosome, while 23% were *trans* PPIs (Supplementary Fig. 9 and Supplementary Data 10). Among these *cis* PPIs, the most prevalent orientation was tandem PPIs (51%), characterized by genes with PPIs being transcribed in the same direction. Divergent PPIs accounted for 40% and convergent PPIs constitute 9%.

Previous chromatin linkage studies in humans suggested that binding sites of transcription factors (TFs) may be involved in chromatin interactions in regulating gene expressions[30]. Therefore, we examined extensive ChIP-seq datasets for experimentally identified TF binding sites[31] and plotted our CAP-C chromatin contacts across these specific TF binding sites. We found that PPIs were highly enriched at these TF binding sites (Fig. 5c). Notably, these enrichments occurred for most transcription factors such as the well-documented TF families bHLH, NAC, and WRKY (Fig. 5d and Supplementary Data 4), suggesting that these primed PPIs preferentially formed at the TF binding sites that may habor these transcription factors.

## Primed promoter-promoter interactions may facilitate the co-regulation of gene expression in response to cold

To test the notion that primed PPIs may be involved in co-regulating gene expression, we performed the co-expression analysis for all three conditions[32]. We calculated the Pearson correlation coefficient for expression of all gene pairs involved in PPIs. Randomly selected gene pairs with the same number and similar distance distribution of the PPI connected gene pairs were built as control groups. We found that the Pearson correlation coefficients for expression of all gene pairs with PPIs for all three conditions were significantly higher than those from the random gene pairs (Fig. 6a and Supplementary Fig. 11), indicating that the gene pairs with PPIs tend to be co-expressed. Thus, our results suggest that the primed PPIs may contribute to the co-regulations of gene expression in response to cold treatments.

Notably, among these gene pairs, there was one primed PPI hotspot where the promoter of the rDNA gene (*AT3G41768*) interacted with promoters of 313 genes. Interestingly, the 313 genes and the rDNA gene were co-downregulated after 3 h of cold treatment and co-upregulated after 12 h of cold treatment (Fig. 6b). This co-regulation of gene expression might assist the extensive RNA-mediated chromatin interactions in these loci[26]. Thus, our results suggested that this primed PPI network comprising the promoters of the rDNA gene and other 313 genes together formed the transcription factories for co-regulating gene expression in response to cold.

Further assessment of other co-regulatory gene pairs containing primed PPIs showed that for many gene pairs where one gene was involved in stress response, its co-regulatory gene partner was mainly involved in either cellular metabolic processes or developmental processes (Supplementary Data 6). To better visualize PPI-mediated co-regulation of gene expression, we plotted the primed PPIs in examples of gene pairs. In Fig. 6c, we illustrate three co-differentially expressed gene pairs with primed PPIs: (1) *AT1G60220* and *AT1G60230* encode a stress-responsive SUMO protease and a SAM superfamily protein which participates in chloroplast function, respectively. We found their expression levels were co-downregulated after 3 h of cold treatment and co-upregulated after 12 h of cold treatment (Fig. 6c). (2) *AT1G78210* and *AT1G78230* respectively encode alpha/beta-Hydrolases for enhancing tolerance to freezing and the outer arm dynein light chain 1 (ODALC1) protein that regulates intracellular

transport, cell division, and organelle positioning. We found both genes were co-upregulated after 3 h of cold treatment and recovered back to control levels after 12 h of cold treatment. (3) *AT1G80940* and *AT1G80950* encode a SNF1-RELATED PROTEIN KINASE that regulates stomatal closure in response to stress and LYSOPHO-SPHATIDYLETHANOLAMINE involved in cell membrane metabolism, respectively. Both gene expression levels were strongly suppressed after 3 h of cold treatment and returned to control levels after 12 h of cold treatment.

We further assessed the T-DNA inserted mutants where a previous study suggested that T-DNA insertion is capable of disrupting chromatin interactions[33]. We identified a T-DNA insertion mutant, the *lpeat1-1* mutant (SALK_065009), with the T-DNA inserted into the promoter of gene *AT1G80950* across the PPI sites (Supplementary Fig. 12a). We then confirmed the PPIs detected in our CAP-C in the wild-type plants using 3C and reciprocal 3C experiments[14,34] (Supplementary Fig. 12b, c). The PPIs between the gene pair *AT1G80940* and *AT1G80950* were disrupted in the T-DNA insertion mutant (Supplementary Fig. 12b, c). Similarly, we confirmed another two T-DNA inserted mutants: the *oadlc-1* (SALK_093069) (Supplementary Fig. 12d) and the *rssp-1* (SALK_093069) (Supplementary Fig. 12g). The identified PPIs between the gene pair *AT1G78210* and *AT1G78230*, and PPIs between the gene pair *AT1G60220* and *AT1G60230* were also disrupted in these T-DNA insertion mutants (Supplementary Fig. 12e, f, h, i).

Following the validations of abolished PPIs in these T-DNA mutants, we further measured the co-regulation of gene expressions. In the comparison of 3-h 4 °C vs. 22 °C in the Col-0 wild-type plants, a distinct co-downregulation trend was observed in the gene pair *AT1G80940* and *AT1G80950*, consistent with the results from the plaNET-seq data (Fig. 6 and Supplementary Fig. 13a, b). Similarly, in the comparison of 12-h 4 °C vs. 3-h 4 °C, a co-upregulation pattern was observed (Fig. 6 and Supplementary Fig. 13a, b). In the *lpeat1-1* mutant, the gene expression of *AT1G80950* was significantly reduced at 22 °C along with significantly different cold responses (Supplementary Fig. 13b). Interestingly, in the *lpeat1-1* mutant, the expression of the PPI paired gene *AT1G80940* was also significantly decreased at 22 °C (Supplementary Fig. 13b). The corresponding plant response expression patterns under different cold treatments were very different from those observed in the wild-type plants (Supplementary Fig. 13b). Similar results were observed in the other two T-DNA mutants (Supplementary Fig. 13c–f). To further validate the effect of PPI disruption, we also successfully generated the CRISPR/Cas9 stable transgenic lines for the two sets of gene pairs connected by PPIs. For the gene pair *AT1G80940* and *AT1G80950* connected by PPIs, we obtained a CRISPR/Cas9 stable transgenic line with a 512-bp region within the promoter of *AT1G80940* across the PPI sites (Supplementary Fig. 14a). This deletion was confirmed by gel analysis and Sanger sequencing (Supplementary Fig. 14b, c). In the context of the promoter deletion mutants for *AT1G80940*, the expression patterns of both genes in plant response to the different cold treatments were very different from those in the wild-type plants (Supplementary Fig. 14d). Similarly, we have also validated our identified PPIs in the gene pair of *AT1G78210* and *AT1G78230* (Supplementary Fig. 14e). For the gene pair *AT1G78210* and *AT1G78230* connected by PPIs, guide RNAs were designed to target and delete a 565-bp region within the promoter of *AT1G78210* across the PPI sites (Supplementary Fig. 14f, g). In the promoter deletion mutants for *AT1G78210*, the expression patterns of both genes in response to cold were also dramatically different from those in the wild-type plants (Supplementary Fig. 14h). The significant impacts on the co-regulatory gene expression we observed further underscore the importance of PPIs in co-regulating gene expressions. Taken together, these results indicated that PPIs are important for the gene expressions of these gene pairs and their co-regulations in response to cold.

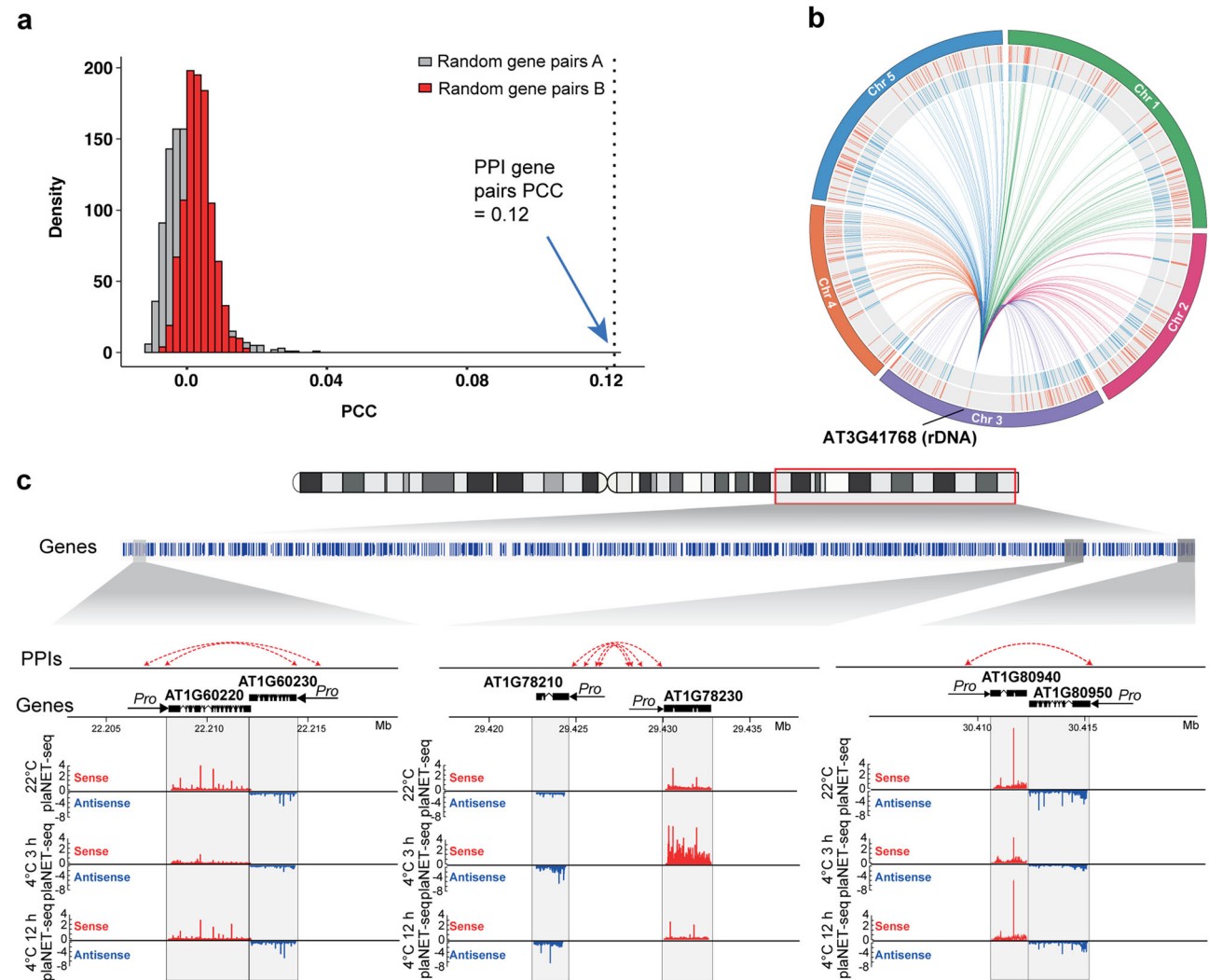

**Fig. 6 | Promoter-promoter interactions may facilitate co-regulation of gene expression in response to cold.** **a** Pearson correlation coefficients of expressions from random gene pairs and PPI connected gene pairs at 22 °C. The Pearson correlation coefficients for gene pairs connected by promoter-promoter interaction (PPI) are significantly higher compared to those for randomly selected genes pairs. Randomly selected gene pairs with the same number and similar distance distribution of the PPI connected gene pairs were built as control groups. Random gene pairs A: random gene pairs with the same number; random gene pairs B: random gene pairs with similar distance distribution of PPI gene pairs. Both random

procedures were repeated 1000 times. $p < 0.0001$, two-sided Spearman's rank correlation. **b** A circus plot showed the interactions between the promoters of *AT3G41768* (rDNA) and 313 genes in the inner circle. From inner to outside, the second circle (blue) showed the co-downregulation of gene expression between 22 °C and 3-h cold treatment (4 °C). The third circle (red) showed co-upregulation of gene expression between 3-h and 12-h cold treatments (4 °C). The outermost circle displayed the five *Arabidopsis* chromosomes. **c** Examples showed PPIs of each gene pair. plaNET-seq profile[2] showed the transcriptional level of each gene locus at 22 °C, 3-h, and 12-h cold treatments (4 °C).

Taken together, our validations from both T-DNA mutants and CRISPR-Cas9-based deletion mutants indicate the importance of PPIs in the co-regulations of gene pair expression in response to cold stress.

## Discussion

Recent effort has improved our understanding of chromatin architectures in plants[35]. Whether plant chromatin conformations have a regulatory impact on gene expression remains largely unknown. As sessile organisms, plants deploy global transcription reprogramming in response to environmental changes such as cold stress[1]. It is unclear whether chromatin conformations alter in response to cold stress. If conformational changes occur, are these chromatin dynamics responsive to transcriptional reprogramming? Here, we applied CAP-C methodology in *Arabidopsis* to chemically crosslink proximal genomic DNA loci directly. We generated the first high-resolution chromatin contact map and identified fine-scale chromatin features that were associated with transcription dynamics, contributing to plant response

to the cold. We also uncovered comprehensive PPI networks prewired in the genome, that may facilitate cold-responsive gene expression co-regulation.

Our study successfully generated the first high-resolution chromatin contact map with fine-scale details of ~200 base pairs in *Arabidopsis* (Supplementary Fig. 1c). The use of direct DNA crosslinking with bridge linker ligation has enabled us to detect short-range chromatin contacts with low background noise, resulting in high accuracy and sensitivity (Figs. 1 and 2 and Supplementary Figs. 1 and 2). Our CAP-C approach had a distinct advantage over traditional Hi-C as it reduced reliance on protein-DNA crosslinking. Thus, we were able to capture strong chromatin contacts at centromeric regions that were not observed in Hi-C (Fig. 1b), consistent with the DNA FISH fluorescent pattern[10]. As a result, CAP-C uncovered intrinsic chromatin conformation features with high reproducibility (Supplementary Fig. 1b). Using CAP-C, we identified well-established chromatin contacts at the gene level (Fig. 2a, b) as well as E-P interactions between active

enhancers and neighboring gene promoters (Fig. 2c, d). Notably, our CAP-C detected E-P chromatin contacts on enhancer-transcribed RNA (eRNAs)[22,25,36], further supporting CAP-C's sensitivity for identifying local *cis*-regulatory elements. Therefore, we found CAP-C to be well suited for discovering intrinsic local chromatin conformations and Hi-C better suited for exploring protein-mediated long-range chromatin interactions. Combining both approaches could be applied to systematically decipher chromatin features at different scales.

Our high-resolution chromatin contact map enabled us to reveal that fine-scale chromatin contacts were responsive to transcriptional activities (Fig. 3a) at the gene level. High chromatin contacts positively correlated with active histone modifications and negatively correlated with repressive histone modifications (Fig. 3b), similar to observations in mammals[18]. Additionally, the association between chromatin accessibility and chromatin interactions (Supplementary Fig. 5) may indicate the gene folding between TSS and the TTS[37]. Besides transcription-associated enrichment of chromatin contacts around transcription start and termination sites (Fig. 3c), we found significant positive correlations between Pol II pausing and local chromatin contacts (Fig. 3d and Supplementary Fig. 7a), indicating a potential link between these two features. Moreover, our results indicated that genes with high Pol II densities across exon-intron junction sites exhibited more chromatin contacts than those with low Pol II densities (Fig. 3e). Notably, this disparity was most prominent at the first exon-intron junction, which is critical for Pol II elongation activity (Fig. 3f), suggesting that local chromatin conformations may be finely primed across the gene body to define transcription activity.

Interestingly, we found these local chromatin conformations changed dynamically in response to cold, consistent with changes in Pol II pausing signals, particularly downstream of the 5′SS (Fig. 4). The Pol II pausing signal downstream of the 5′SS is an important checkpoint during Pol II elongation[2], thus affecting transcription activity. Therefore, being able to dynamically tune local chromatin conformations inside the gene may offer a novel way of regulating cold-responsive transcription activity to enhance plant thermo-resilience to environmental temperature stress.

In contrast to the plasticity of local chromatin conformations, we did not find large-scale chromatin conformations altered in response to cold (Supplementary Fig. 8). Instead, we found a large thermo-stable PPI network across all five chromosomes that is likely primed in the genome (Fig. 5a, b). Notably, ChIP-seq experimental data showed these PPI sites were highly enriched across TF binding sites, (Supplementary Data 2 and Fig. 5c, d), suggesting these PPIs are primed across the genome that may harbor transcription factors. We also found that the gene pairs with PPIs tend to be co-expressed in three conditions (Fig. 6a), implying the PPIs may participate in the co-regulations of gene expression in response to cold treatments. Due to chromatin interactions enabling the connectivity between two genes, certain transcription factors may bind with these interacting PPI sites and co-regulate the gene pair. For instance, the PPI sites of the three gene pairs in Fig. 6c were across the binding sites of a cold-responsive transcription factor ICE1 (*Inducer of CBF Expression 1*)[31], suggesting that these PPIs may harbor ICE1 for co-regulating gene expression in plant response to cold. This hypothesis needs to be further proved in future work. Notably, those PPI-mediated co-regulated genes are not only associated with stress response functions, but also cellular metabolites and developmental processes (Supplementary Data 6 and 11), suggesting that PPI may contribute to balancing the trade-off between plant development and stress response. The principle of trade-off in plants underscores a strategic allocation of resources, directing plants toward either a stress response or growth contingent upon prevailing conditions. The trade-off mechanism exemplifies how resource allocation is dynamically modulated to enhance plant adaptability and fitness, even amid challenging environmental circumstances[38,39]. The primed PPIs captured in our CAP-C library have revealed a diverse array of pathways, particularly noteworthy were pathways associated with stress responses, anatomical structure development, biosynthesis, and metabolic pathways. Within these categories, we identified a delicate interplay of genes that function as positive and negative regulators within their respective pathways. The co-regulation of these genes could possibly be a mechanism for the trade-off phenomenon.

In plant cold-responsiveness, we observed dynamic changes of local chromatin conformations contributing to the fine-tuning of transcription activities, in contrast to the large-scale chromatin conformations which remained largely stable, possibly due to maintaining global genome integrity in cold. A prior Hi-C investigation revealed that rice chromosomal configurations underwent significant changes at a large scale under the cold treatment at 16 °C, resulting in a reduction of long-range interactions exceeding 1 Mb and an increase in A-B compartment interactions[40]. Another previous Hi-C study of heat stress in metazoans reported a similar observation that global chromatin architectures remain unchanged while local distal chromatin interactions were relatively plastic[41]. A recent Hi-C study on tomato chromatin organization reported that heat stress strongly affected global chromatin reorganization and the Heat Shock Factor (HSF) transcription factor HSFA1a was required for chromatin reorganization[42]. Our observation of the unchanged large-scale chromatin conformations in response to cold may be due to the unique chromatin architecture of *Arabidopsis*, which is topologically associated domain (TAD) deficient or thermodynamically stable under conditions at 4 °C that preferably maintain genome integrity. It is also possible that plants adopt different strategies in response to different stresses.

In summary, our study generated the first fine-scale *Arabidopsis* intrinsic chromatin landscape with high resolution and low background noise, facilitating both the determination of local chromatin contacts at the gene level as well as an extensive, globally primed PPI network. Our successful demonstration of CAP-C application in *Arabidopsis* should stimulate future exploration of intrinsic fine-scale chromatin architecture, particularly local chromatin conformations in other plant species. Our findings of cold-responsive local chromatin conformation-associated transcriptional dynamics open a novel pathway for investigating plant stress-responsive regulatory transcription. Our primed PPI network across the genome may be involved in balancing the trade-off between plant development and stress response.

## Methods

### Plants and growth conditions

The *Arabidopsis thaliana* ecotype Columbia (Col-0) and T-DNA insertion mutants (SALK_065009, SALK_093069, SALK_126003) were obtained from the Nottingham Arabidopsis Stock Centre (NASC), *Arabidopsis thaliana* ecotype Columbia (Col-0) seeds were sterilized using 70% ethanol for 10 min, washed with distilled water three times, and plated on one-half strength Murashige and Skoog medium supplemented with 1% sucrose. Plates were stratified for 4 days at 4 °C and then moved to long-day conditions with 16 h light/8 h dark for 10 days. The seedlings were then harvested and fixed for CAP-C library construction. The cold treatment was performed as described[2].

### The CAP-C library construction

Seedlings were crosslinked in 1% formaldehyde for 15 min under a vacuum and kept on ice throughout the process. Crosslinking was then quenched with 0.125 M glycine for 5 min under a vacuum. The seedling tissues were rinsed three times and dried with paper towels, fast-frozen, and stored at −80 °C for nuclei isolation later. To start nuclei isolation, frozen tissue was ground into a fine powder, and Honda buffer (20 mM HEPES, 0.44 M sucrose, 1.25% Ficoll 400, 2.5% Dextran T40, 10 mM MgCl$_2$, 0.5% Triton X-100, 5 mM Dithiothreitol (DTT), and 1× Protease inhibitor cocktail) added. The homogenate was then

centrifuged at $3000 \times g$ under 4 °C and washed several times at $2000 \times g$ until the pellet turned gray. The resulting pellet was resuspended in 1 ml CAP-C buffer (50 mM Tris·HCl pH 7.5, 100 mM NaCl, 1% Triton X-100, 1 mM $MgCl_2$, 0.1 mM $CaCl_2$, 1× protease inhibitor mixture). Purified nuclei were then photo-crosslinked with dendrimer by irradiating under 365 nm UV light for 30 min. Crosslinked chromatin was then digested with proteinase K overnight. After purification with phenol/chloroform, the digested DNA-Dendrimer complexes were then fragmented by DNaseI into 50–200 bp fragments. The bridge linker (customized with IDT: F: /5Phos/GTCAGA/iDBCON/AAGA-TATCGCGT, R:/5Phos/CGCGATATC/iBiodT/TATCTGACT) was then added to the reaction at 37 °C for 2 h to allow for the click reaction with the Et3N branch of the dendrimer. After the removal of the free adapters, ligation was performed between the DNA-dendrimer-adapter complex by T4 DNA ligase. Chimeric fragments were then pulled down by the biotin label on the adapter. Sequencing libraries were constructed using the NEB Library prep kit. Libraries were then subjected to the MGI platform (BGI) for 150 bp pair-end sequencing.

### The CAP-C data processing
The 150 bp pair-end FASTQ formatted files were aligned to the TAIR10 reference genome using HiC-Pro[43] (version 3.0.0). The linker sequence was trimmed by cutadapt[44] (version 1.18). R1 and R2 were merged according to the previous method[7]. The mapping result was filtered using MAPQ ≥ 10 for each sample. Juicer[45] was then used to convert the merged-sorted file into a.hic formatted file. All contact matrices at the 200 bp resolution were visually represented as contact maps using HiCExplorer[46] (version 3.5.1) and Genomic-Interactive-Visualization-Engine (GIVE)[47]. Genome-wide contact decaying was shown using hicPlotDistVsCounts from HiCExplorer[46]. Only valid and filtered contact pairs were used for all downstream analyses. The characterization of CAP-C mapped reads of all replicates from all growth conditions used in this study was summarized in Supplementary Data 1.

A pre-formatted text file was prepared before converting it into a.hic formatted file with the following resolutions (200 bp, 2 Kb, 10 Kb, 20 Kb). To generate a CAP-C merge file, G3 replicated libraries in the pre-formatted text file were concatenated and merged-sorted. Juicer[45] was then used for converting the merged-sorted file into a.hic formatted file. All contact matrices visually represented as contact maps were VC-normalized with Juicer[45]. The reproducibility between two biological replicates was calculated using hicCorrelate from HiCExplorer[46].

### The stable chromatin contacts calling
Genome-wide stable chromatin contacts were identified using Fit-Hi-C[15,48] (version 2.0.7) at the 200 bp resolution, following a previously reported method[49] (Supplementary Data 1 and 7).

### The promoter–promoter network illustration
The promoter-promoter interaction network was integrated into a global network using Cytoscape (version 3.9.1)[50] as shown in Fig. 5a, the network was visualized by the third-party application AllegroLayout. Gephi[51] (version 0.10.0) with ForceAtlas 2 was used for a closer look visualization of the promoter-promoter interaction network.

### ChIP-seq and ATAC-seq data processing
ChIP-seq datasets for different histone modifications[31,52] were processed using the reported pipeline[53]. ATAC-seq data were processed according to the previous report[54]. We followed the methodology used by the authors from which the data originated. The downloaded data were removed adapters by using trim_galore. The cleaned data were mapped to *Arabidopsis thaliana* reference genome (TAIR10) by using bowtie, with parameters: -X 2000 -m 1. Samtools were used to remove duplicate reads, and reads located on the mitochondria and chloroplasts were removed.

### plaNET-seq, pNET-seq and GRO-seq data processing
plaNET-seq datasets were processed following the reported method[2]. GRO-seq and pNET-seq datasets were processed based on a previously reported study[22]. Calculation of differentially expressed genes was performed using htseq-count[55] and DESeq2[56]. Htseq-count was run with the default parameters and the addition of –stranded=yes.

### 3C quantification, normalization and controls
3C experiment was done by following the previous study[14]. Relative interaction frequencies were determined by qPCR. Normalization for differences in DNA concentration among samples was achieved through a loading control (LC) employing primer sets that do not span any restriction site. To address variations in primer efficiency during PCR, normalization was performed against a control template (CT) DNA, which contained all potential ligation products in equimolar proportions. The CT DNA was produced by digesting BAC plasmids with ApoI and subsequent random ligation without dilution. All figures presented in this study reflect the mean values from a minimum of two distinct biological samples. Each 3C DNA preparation underwent quantification through independent qPCR assays, and each qPCR experiment consisted of triplicate analyses for all DNA samples. The complete list of primers and BAC plasmids employed in the experiment can be found in Supplementary Data 12.

### Generation of CRISPR mutants
Specific sgRNAs were chosen for the generation of CRISPR/Cas9 constructs and were subsequently cloned into the pICSL002217A plasmid (TSLSynBio). A comprehensive list of sgRNA primers can be found in Supplementary Data 12.

### Quantitative real-time PCR
RNA was extracted using RNeasy Plant Mini Kit. First-strand cDNA was synthesized using reverse transcriptase Superscript III (Invitrogen) and oligo dT primer. Quantitative qRT-PCR was performed with Light-Cycler® 480 SYBR Green I Master (Roche) using CFX96 Touch Real-Time PCR Detection System (BIORAD) according to the manufacturer's protocol. *ACTIN2* was used as the internal control. All Primers used in the experiment are listed in Supplementary Data 12.

### Reporting summary
Further information on research design is available in the Nature Portfolio Reporting Summary linked to this article.

## Data availability
Sequencing data in this study were deposited in the Sequence Read Archive (SRA) (https://www.ncbi.nlm.nih.gov/sra) under BioProject ID number PRJNA912614. A full list of primers is available in Supplementary Table 12. Source data are provided with this paper.

## Code availability
The code used for analyses is available at https://github.com/Huakun-Lab/CAP-C-in-Arabidopsis.

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

## Acknowledgements

The study is supported by the National Key R&D Program of China (2023ZD04073), the National Key Research and Development Program of China (2021YFF1000900), the National Natural Science Foundation of China (32170229 and 32000178), the Fundamental Research Funds for the Central Universities (2412023YQ005), the United Kingdom Biotechnology and Biological Sciences Research Council (BBSRC: BB/X01102X/1, BBS/E/J/000PR9788, BB/L025000/1, and BB/N022572/1), and the European Research Council (ERC: 680324). We thank Dr Yilin Xie (Guangzhou) for providing us with the active enhancer data derived from the Arabidopsis pNET-seq data. We thank Dr. Sebastian Marquardt (Copenhagen Plant Science Centre) for his support in analyzing the plaNET-seq data. We thank Dame Professor Caroline Dean (John Innes Centre) for her discussions regarding this work. This research was supported by the Key Laboratory of Molecular Epigenetics' High-Performance Computing Cluster.

## Author contributions

Y.D., C.H. and H.Z. conceived the study; Y.Z., Q.D., Q.L., Q.Y., Y.D., C.H. and H.Z. designed the study; Y.Z., H.Z., Z.W., Q.L., W.S., Q.Y. and L.D. performed the experiments; Y.Z., H.Z., Q.D., H.Y. and J.C. conducted the data analyses; Y.D., C.H., X.C. and H.Z. supervised the analyses; Y.Z. and H.Z. wrote the paper with input from all authors.

## Competing interests

The authors declare no competing interests.
