## [Peer Review File · Nature Communications]

A fine-scale Arabidopsis chromatin landscape reveals chromatin conformation-associated transcriptional dynamicsReviewer #1 (Remarks to the Author):

In the manuscript entitled "A fine-scale Arabidopsis chromatin landscape reveals chromatin conformation-associated transcriptional dynamics", the authors utilized a chemical-crosslinking assisted proximity capture (CAP-C) method to generate a high-resolution chromatin contact map in Arabidopsis, enabling the identification of chromatin interactions at the gene level. Subsequent analysis of these local interactions revealed their association with Pol II activities, which were dynamically reprogrammed in response to cold stress. PPI network identified by CAP-C showed co-regulation of the interactors during cold treatment. This study represents the first application of the CAP-C method in plants and provides valuable insights into fine-scale chromatin interactions. Nevertheless, I have the following main concerns:

The terms "chromatin loop" and "chromatin contacts" are not clearly defined. In this manuscript, it appears that "chromatin loop" refers to chromatin contacts with statistical significance (according to line 447 in the method section). However, I could not find any information on "chromatin loops", such as the number of loops and the distance range of loops, furthermore, whether the statistical modeling of FIT-HiC is suitable to call loops at 200-bp resolution is not justified. On the other hand, the term "chromatin contacts" is used to describe "observed CAP-C reads" (among which the vast majority of sequencing reads come from stochastic chromatin contacts). These two concepts are used interchangeably without giving any reasons, for instance, Figs 3 and 4 are based on normalized CAP-C reads; while Fig 5 and Fig6 are likely based on chromatin loops. Fig6c makes me confused: it depicts both intra- and inter-chromosomal PPI. Were they "loops" identified by FIT-HiC? If yes, I don't think 200-bp resolution works for inter-chromosomal or long-distance intra-chromosomal (e.g., several Mbs) contact calling; if the answer is no, does it mean that PPIs were only observed CAP-C reads, which can be stochastic?

Other comments:

Fig2a and 2b illustrated some short chromatin loops (<500 bp or so). In such a distance range, I am curious to know how CAP-C reads were analyzed so that one can differentiate ligation products from unligated genomic DNA.

The y-axis values of some plots are inconsistent: Fig3e,3f, 4d, 4e have the same y-axis and these panels describe "normalized interactions per bin" over exon/intron junctions. But the values in fig3 (between 1.5 and 1.7) are much lower than those from fig4 (between 0 and 10). Furthermore, the curves in Fig4d and 4e have negative values, by y-axis definition, this is not possible.

Fig2e and 2f have chromatin interactions that I could not understand well. In both panels, there are way more interactions within the locus of interest; on the contrary, the contacts between the locus of interest and its flanking regions appear extremely sparse. Are such extraordinarily strong chromatin contacts within gene body representative?

From lines 110 to 156, the authors clarified that CAP-C can identify local chromatin organizational features, including interactions between individual gene loci, enhancers and promoters, and promoter-promoter interactions. While two examples were presented in Fig2 for each feature, they were not sufficient to illustrate interactions on a genome-wide scale. It would be beneficial to provide genome-wide information on chromatin contacts, such as how many loops were identified, the numbers of E-P and P-P interactions, and so on. Furthermore, systematic characterization of P-P and E-P contacts would be appreciated.

Concerning the association between local chromatin contacts and Pol II activities, the authors claimed that "the regions with high Ser5P Pol II densities exhibited more chromatin contacts than those with low Ser5P Pol II densities (Fig. 3e). Notably, this disparity was most prominent at the first exon-intron junction, which is critical to Pol II elongation activity (Fig. 3f)" (line 193 to 195), in which the signal of Ser 5p Pol II peaked across the 5'SS or the 1st 5'SS regions. In references 2 and 20, which discuss Pol II activities, splicing intermediate within the spliceosome complex that coprecipitated with Ser 5P Pol II at 5'SS were reported, indicating that the signals at 5'SS contain both nascent RNA and splicing intermediates. Could the presence of splicing intermediates mislead the accurate signal of nascent RNAPII transcription? Similarly, in the subsequent section that demonstrates the dynamics of chromatin conformation-associated Pol II activities during cold

response (Fig4f,g and Supplementary Fig. 4b,c), the authors should consider excluding possible effects of splicing kinetics during cold treatment.

In Fig6a and 6b, it is striking that promoter-promoter interacting genes were exclusively co-upregulated or co-downregulated during cold treatment. I am skeptical that not all promoter-promoter interacting gene pairs are present in the plots, instead, it is clear that a filter has been applied so that only a subset of gene pairs are shown. Besides, perhaps "22C or 3-hour cold treatment" (line 701) should be "3-hour vs. 22C", and "3-hour or 12-hour" (line 703) should be "3-hour vs. 12-hour". Further analysis of these genes "showed that for many gene pairs where one gene was involved in stress response, its co-regulatory gene partner was mainly involved in either cellular metabolic processes or developmental processes" (line279 to 281). Generally, stress responses are considered antagonistic to plant growth and development. The authors should discuss the potential reasons behind this observation.

Reviewer #2 (Remarks to the Author):

I don't understand why this manuscript could be considered for reviewed in a nature journal. The Cap-C method has already been published in nature biotechnology. This is just applying it to plant? Is that acceptable now?

I also don't understand why the author would choose Arabidopsis, which has a small genome and no meaningful long-range chromatin interactions. Yes, Cap-C has very good resolution, which is very useful for large animal genome, and this has been nicely demonstrated in the first nature biotechnology paper. But plants, Arabidopsis in particular do not have such chromatin loops. That's why the author couldn't find anything meaningful interaction from their Cap-C at this super high resolution. They then turned to making a "promoter-promoter-interaction" (PIP) network and "transcription-initiation-cluster" (TIC). But those are just highly expressed genes in the transcription factories that got crosslinked together. These "interaction" has no regulatory function. Why would anyone bother to examine those?

The last sentence of the abstract said "...contributed to transcriptional reprogramming, enhancing our understanding of chromatin conformation-associated gene regulation". None of the result in this paper can demonstrate that such chromatin interaction has any regulatory function.

There are too many papers describing this type of chromatin interaction in plants. But no one can confidently say that those interaction has any regulatory function. Very few exceptional cases in some large plant genome, in which a few real enhancers can be positioned hundreds of kb away from the target gene, like b1 and tb1. Those rare cases have been extensively studied and people have published those many years ago. Now, if anyone wants to claim that they find new long-range chromatin interaction that can control gene expression, they can't just show there is an interaction by sequencing. because adjacent accessible chromatin or transcribed genes can be cross-linked in such experiment and detected as if they are interacting. if we want to make such functional claim, we must produce evidence. So for, no one is able to do so. lots of people tried to use CRISPR to remove those so-call distal enhancers" and found no meaningful effect on gene expression. That's why we wont consider the plant chromatin interaction would have the same function as those in the animal nucleus.

We thank the reviewers for their time and effort in helping us improve our manuscript. We have conducted additional new experiments and analyses to strengthen our conclusions. In total, we have added additional 5 supplementary figures and 6 supplementary data into our revised manuscript, accordingly. Please find the detailed responses as follows.

REVIEWER COMMENTS

Reviewer #1 (Remarks to the Author):

In the manuscript entitled “A fine-scale Arabidopsis chromatin landscape reveals chromatin conformation-associated transcriptional dynamics”, the authors utilized a chemical-crosslinking assisted proximity capture (CAP-C) method to generate a high-resolution chromatin contact map in Arabidopsis, enabling the identification of chromatin interactions at the gene level. Subsequent analysis of these local interactions revealed their association with Pol II activities, which were dynamically reprogrammed in response to cold stress. PPI network identified by CAP-C showed co-regulation of the interactors during cold treatment. This study represents the first application of the CAP-C method in plants and provides valuable insights into fine-scale chromatin interactions.

General Response: We thank the Reviewer for their encouraging feedback: “*This study represents the first application of the CAP-C method in plants and provides valuable insights into fine-scale chromatin interactions*”. We are also very grateful for their suggestions to help us improve our manuscript. During our revision, we performed additional new analyses and experiments that fully addressed all the comments.

Nevertheless, I have the following main concerns: The terms “chromatin loop” and “chromatin contacts” are not clearly defined. In this manuscript, it appears that “chromatin loop” refers to chromatin contacts with statistical significance (according to line 447 in the method section). However, I could not find any information on “chromatin loops”, such as the number of loops and the distance range of loops, furthermore, whether the statistical modeling of FIT-HiC is suitable to call loops at 200-bp resolution is not justified.

Response: We thank the Reviewer for the comments. We apologize for the confusing terms and have clarified our manuscript as follows.

CAP-C's unique advances include a bi-functional bridge linker, attached to dendrimers via "Click Chemistry"¹. This linker connects proximal contacts on the same dendrimer, reducing random DNA collisions, minimizing unligated genomic DNA ligation products, and enhancing specificity in identifying chromatin proximities. Consequently, only DNA fragments crosslinked and reacted with the linker are captured in our libraries. **We termed the read pairs that contain the bridge linker in the middle as the valid "chromatin contacts"**. Furthermore, we assessed the reproducibility of chromatin contacts in our CAP-C libraries. We found that the reproducibility is very high between two biological replicates (**Supplementary Figure 1b, Rebuttal Table 1 and Rebuttal Figure 1**). We illustrated the reproducibility with both zoom-in regions and chromosome-wide measurements (**Supplementary Figure 1b, Rebuttal Table 1 and Rebuttal Figure 1**). The high reproducibility shown supports the advantages of CAP-C on the specificity and sensitivity in measuring the chromatin interactions.

Furthermore, to facilitate comparison with previous studies^{2,3,4} in **Figure 2**, we only used Fit-Hi-C (the latest reimplemented version FitHiC2⁵) to identify highly enriched chromatin contacts (termed as **stable chromatin contacts**). The Fit-Hi-C computes accurate empirical null models of stable chromatin contact probability without any distribution assumption, corrects for binning artifacts and identifies chromatin contacts with the high confidence. Thus, it is capable of identifying highly enriched chromatin interactions (**stable chromatin contacts**) with different distances^{5,6}. As shown in **Figure 2**, these stable chromatin contacts identified from our CAP-C, were used to be compared with previous chromatin conformation studies^{2,3}. We have provided new supplementary information on the number and the distance range of these stable chromatin contacts (**New Supplementary Data 7**).

Rebuttal Table1. HiCRep⁷ reproducibility at 200-bp resolution (measured by the stratum-adjusted correlation coefficient).

	Chr1	Chr2	Chr3	Chr4	Chr5
CAP-C 22°C Rep1 vs Rep2	0.9461	0.9276	0.9421	0.9427	0.9299
CAP-C 4°C 3h Rep1 vs Rep2	0.9518	0.9536	0.9479	0.9526	0.9494
CAP-C 4°C 12h Rep1 vs Rep2	0.9735	0.9713	0.9703	0.9736	0.9717

Rebuttal Figure 1. High reproducibility of chromatin contacts across the genome.

Direct comparisons of chromatin contacts between two biological replicates at 22°C in our CAP-C libraries were made using Integrative Genomic Viewer⁸. Chromatin contacts from each replicate were shown in the upper and lower halves as arcs. Five zoom-in regions from different chromosomes are shown to exemplify the reproducibility of chromatin contacts in each replicate. Each line in the figure represents a chromatin contact for the respective region.

On the other hand, the term “chromatin contacts” is used to describe “observed CAP-C reads” (among which the vast majority of sequencing reads come from stochastic chromatin contacts). These two concepts are used interchangeably without giving any reasons, for instance, Figs 3 and 4 are based on normalized CAP-C reads; while Fig 5 and Fig6 are likely based on chromatin

loops. Fig6c makes me confused: it depicts both intra- and inter-chromosomal PPI. Were they “loops” identified by FIT-HiC? If yes, I don’t think 200-bp resolution works for inter-chromosomal or long-distance intra-chromosomal (e.g., several Mbs) contact calling; if the answer is no, does it mean that PPIs were only observed CAP-C reads, which can be stochastic? Fig2a and 2b illustrated some short chromatin loops (<500 bp or so). In such a distance range, I am curious to know how CAP-C reads were analyzed so that one can differentiate ligation products from unligated genomic DNA.

Response: We only used Fit-HiC in validating our CAP-C with previous studies^{2,3,4} on individual genes. For other analyses, we directly characterized our high-resolution accurate chromatin contacts in the following figures (**Figures 3-6**). **Figure 5 and 6** are based on the chromatin contacts. **Figure 6c** displays both *trans* and *cis* chromatin contacts for the gene *AT3G41768*. These PPIs were highly reproducible between the two biological replicates (**Rebuttal Figure 2**). Thus, our identified PPIs are not stochastic.

Our CAP-C procedures were designed to remove the unligated genomic DNAs⁹ where all the valid read pairs (chromatin contacts) contain the bi-functional bridge linker. In **Figure 2a** and **2b**, we specifically enriched the stable chromatin contacts derived from Fit-Hi-C to compare with the chromatin loops generated from previous low-resolution HiC data. We further assessed the reproducibility of all our observed chromatin contacts across *FLC* loci and *APOLO* and found high reproducibility between two replicates (**Rebuttal Figure 3**), further supporting the specificity and sensitivity of our CAP-C.

Rebuttal Figure 2. High reproducibility of PPIs with *AT3G41768* across the genome.

Circus plots showing the PPIs formed by the promoter of *AT3G41768* (rDNA) across the genome between two replicates in each condition.

Rebuttal Figure 3. High reproducibility of chromatin contacts across the gene locus.

a and **b** Direct comparison of chromatin contacts across *FLC* (**Fig. 2a**) and *PID*&*APOLO* (**Fig. 2b**) in two biological replicates from our CAP-C libraries at 22°C is shown, respectively. Each line represents a valid chromatin contact.

The y-axis values of some plots are inconsistent: Fig3e,3f, 4d, 4e have the same y-axis and these panels describe “normalized interactions per bin” over exon/intron junctions. But the values in

fig3 (between 1.5 and 1.7) are much lower than those from fig4 (between 0 and 10). Furthermore, the curves in Fig4d and 4e have negative values, by y-axis definition, this is not possible.

Response: We thank the Reviewer for the important comments. We apologize for any confusion. We observed a significantly high peak of average chromatin contacts downstream of exon-intron junctions in CAP-C libraries under three conditions (**Rebuttal Figure 4 and Rebuttal Table 2**). The average chromatin contacts across the genomes were slightly different among the CAP-C libraries under three conditions due to different sequencing depths. In order to compare the dynamic changes of this chromatin-contact peak under different conditions, we conducted the Z-score normalization for all three conditions where we observed that the chromatin contacts located downstream of the 5'SS dropped sharply during 3 hours in the cold, but gradually reverted to control levels during 12 hours in the cold (**Fig. 4d and 4e**). We have clarified this in our revised figure legend highlighted in yellow.

Rebuttal Figure 4. The average chromatin contacts across exon-intron junctions under three conditions.

a, b Meta-profiles showing high peaks of chromatin contacts 100 bp upstream and 200 bp downstream of the **(a)** 5' SS or **(b)** the 1st 5' SS under three conditions before Z-score normalization are evident (22°C in red, 4°C treatment for 3h in dark blue, 4°C treatment for 12h in green). The grey shaded area represents the 95% confidence interval. The dashed line indicates the position of the exon-intron junction.

Rebuttal Table 2: Statistical significance assessment for peaks compared to the background in Rebuttal Figure 4

Peak compared to the whole region.

	Average Chromatin Contacts Peak Value	t	df	p value
CAP-C 22°C	1.630166	-12.142	34	6.5E-14
CAP-C 4°C 3h	1.422726	-11.562	34	2.5E-13
CAP-C 4°C 12h	1.446476	-11.623	34	2.2E-13

Peak compared to the region before the peak.

	Average Chromatin Contacts Peak Value	t	df	p value
CAP-C 22°C	1.630166	-10.864	16	8.57E-09
CAP-C 4°C 3h	1.422726	-9.842	16	3.43E-08
CAP-C 4°C 12h	1.446476	-10.363	16	1.67E-08

Peak compared to the region after the peak.

	Average Chromatin Contacts Peak Value	t	df	p value
CAP-C 22°C	1.630166	-12.142	34	6.5E-14
CAP-C 4°C 3h	1.422726	-11.562	34	2.5E-13
CAP-C 4°C 12h	1.446476	-11.623	34	2.2E-13

Fig2e and 2f have chromatin interactions that I could not understand well. In both panels, there are way more interactions within the locus of interest; on the contrary, the contacts between the locus of interest and its flanking regions appear extremely sparse. Are such extraordinarily strong chromatin contacts within gene body representative?

Response: We thank the Reviewer for this comment. **Figure 2e** and **2f** were used to demonstrate that CAP-C not only verified long-distance chromatin contacts previously reported by BL-Hi-C¹⁰

but also emphasised our capability to identify more short-range chromatin contacts (**Rebuttal Figure 5**). In general, high-resolution methods such as our CAP-C⁹, BL-Hi-C¹⁰ and Micro-C¹¹ all tend to capture more gene-body short-distance chromatin interactions as compared to long-distance chromatin interactions. The chromatin contacts in our CAP-C libraries were globally more enriched in the gene-body regions across the genome as compared to their flanking intergenic regions (**Rebuttal Figure 6 and 7**). These gene-body chromatin contacts are likely to be associated with Pol II activities. It is also likely that the CAP-C method tends to capture more short-distance chromatin contacts.

Rebuttal Figure 5. The distance distribution of chromatin contacts of the genes *AT1G58602* (a) and *AT1G01320* (b).

a and **b** The distance distribution of chromatin contacts of genes *AT1G58602* and *AT1G01320*, respectively. The x-axis represents the distance of chromatin contacts, while the y-axis represents the number of chromatin contacts.

Rebuttal Figure 6. Higher enrichment of chromatin contacts within gene-body regions.

Heat plot comparison of chromatin contacts between the gene-body regions and their flanking intergenic regions. It profiles the strength of CAP-C chromatin contacts between the gene-body regions and their 100bp upstream and downstream intergenic regions across the genome. TSS, transcription start site. TTS, transcription termination site. The color scale ranges from blue to red, with red indicating strong chromatin contacts and blue indicating weak chromatin contacts.

Rebuttal Figure 7. Globally higher enrichment of chromatin contacts within gene-body regions.

Chromatin contacts in the gene-body regions are significantly stronger than those in the intergenic regions. (p value $< 2.2e-16$, based on Wilcoxon rank sum exact test)

From lines 110 to 156, the authors clarified that CAP-C can identify local chromatin organizational features, including interactions between individual gene loci, enhancers and promoters, and promoter-promoter interactions. While two examples were presented in Fig2 for each feature, they were not sufficient to illustrate interactions on a genome-wide scale. It would be beneficial to provide genome-wide information on chromatin contacts, such as how many loops were identified, the numbers of E-P and P-P interactions, and so on. Furthermore, systematic characterization of P-P and E-P contacts would be appreciated.

Response: We thank the reviewer for this useful suggestion. We have conducted comprehensive characterizations of E-P and P-P chromatin contacts and provided extensive new supplementary figures and tables in our revised manuscript (**New Supplementary Figures 3-4, New Supplementary Figure 8 and New Supplementary Data 8-11**).

Previous studies showed that enhancers work as transcriptional regulatory elements, orchestrating gene expression from a distance^{12, 13}. We then conducted an analysis on the distribution of the distances for E-P chromatin contacts determined in our CAP-C libraries in all the chromosomes (**Rebuttal Figure 8**). We found that the majority (89%) of E-P chromatin contacts were distal while 11% were proximal, aligning with previous findings^{12, 13}. For both proximal and distal E-P chromatin contacts, there are three orientations: convergent, divergent, and tandem. For both proximal and distal enhancers, the tandem orientations were predominant (**Rebuttal Figure 9**). There is a noticeable increase in the convergent E-P chromatin contacts in the group of distal E-P chromatin contacts in comparison with those in the proximal group. Taken together, our CAP-C detected both proximal and distal E-P chromatin contacts with a predominant tandem orientation. We have added these characterizations of E-P chromatin contacts analysis in our revised manuscript in lines 144-154, highlighted in yellow.

Rebuttal Figure 8 (New Supplementary Figure 3). The distribution of distances for E-P chromatin contacts in all chromosomes.

a-f The distribution of the distances for all E-P chromatin contacts in all five chromosomes was plotted respectively: total E-P chromatin contacts for all chromosomes (**a**) and E-P chromatin contacts in each individual chromosome (**b-f**).

Rebuttal Figure 9 (New Supplementary Figure 4). Proportion of different types of E-P chromatin contacts. The proportion of convergent, divergent, and tandem orientations in both proximal and distal E-P chromatin contacts.

Promoter-promoter interactions (PPIs) play a pivotal role in regulating gene regulation. We conducted a comprehensive analysis of the PPIs captured in our CAP-C libraries. A notable majority (77%) of these interactions occurred in a *cis* manner, where interactions take place on the same chromosome, while 23% were *trans* PPIs (**Rebuttal Figure 10**). Among these *cis* PPIs, the tandem orientation was the most prevalent (51%), characterized by genes with PPIs being transcribed in the same direction. Divergent PPIs accounted for 40% and convergent PPIs constituted 9% (**Rebuttal Figure 10**).

Multiple PPIs form clusters that were primed for regulating gene expression, potentially providing a topological framework for simultaneous transcriptional control¹⁴. We have shown the primed PPI network and PPI nodes in Figure 5a. The total PPI nodes are listed in **Rebuttal Table 3**. Here we also present some PPI nodes in **Rebuttal Figure 11**. For instance, the PPI node centred by the *GONST5* (*GOLGI NUCLEOTIDE SUGAR TRANSPORTER 5*) brought several Golgi-related genes such as ER-Golgi transport (*P24BETA2*, *SKS7*, *SKS8* *UMAMIT19*) with biotic stress-related genes such as *LYMI* via PPIs. The root development gene *TET13* node contains *ATPI4K* *GAMMA 7* (involved in phosphoinositide signalling and nutrient uptake), *RGF8* (Root Meristem Growth Factor), *NMD3* (ribosomal protein subunit export), and the Pentatricopeptide repeat (PPR) Superfamily (potentially linked to pathogen response), forming a co-regulatory network between growth and stress response. Another meristem gene, *WOX8*, interacted with *RAC2*, which is associated with cell cycle, *ATNUDT11* (CoA pyrophosphatase) and *APG7* associated with plant immunity, forming another co-regulatory network. These examples may indicate how plants may

fine-tune the interplay of primed PPIs to enhance adaptability and fitness within diverse environmental contexts.

Rebuttal Figure 10 (New Supplementary Figure 8). Overview of PPIs captured in CAP-C libraries.

a Proportion of *cis* and *trans* PPIs. **b** Proportion of different types of *cis* promoter-promoter interactions. **c-h** The distribution of distances for all *cis* PPIs in all the chromosomes was plotted respectively: total PPIs for all five chromosomes (**c**) and PPIs in the individual chromosome (**d-h**).

Rebuttal Figure 11. Examples of the PPI nodes.

Three PPI nodes were generated using *Gephi*¹⁵.

Concerning the association between local chromatin contacts and Pol II activities, the authors claimed that “the regions with high Ser5P Pol II densities exhibited more chromatin contacts than those with low Ser5P Pol II densities (Fig. 3e). Notably, this disparity was most prominent at the first exon-intron junction, which is critical to Pol II elongation activity (Fig. 3f)” (line 193 to 195), in which the signal of Ser 5p Pol II peaked across the 5’SS or the 1st 5’SS regions. In references 2 and 20, which discuss Pol II activities, splicing intermediate within the spliceosome complex that coprecipitated with Ser 5P Pol II at 5’SS were reported, indicating that the signals at 5’SS contain both nascent RNA and splicing intermediates. Could the presence of splicing intermediates mislead the accurate signal of nascent RNAPII transcription? Similarly, in the subsequent section that demonstrates the dynamics of chromatin conformation-associated Pol II activities during cold response (Fig4f,g and Supplementary Fig. 4b,c), the authors should consider excluding possible effects of splicing kinetics during cold treatment.

Response: We thank the Reviewer for this feedback. Due to the limitation of NET-seq/plaNET-seq methods, splicing intermediates can be detected^{16, 17, 18}, splicing intermediate RNA associated with the spliceosome could co-precipitate with RNA polymerase II, leading to potential inaccuracies in using the 5' and 3' splice site positions to accurately determine the location of nascent Pol II transcription. Thus, in Kindgren’s study¹⁷ they filtered out these ambiguous read positions. When we conducted the data processing steps according to Kindgren’s pipeline, we also filtered those read positions to avoid uncertainty (https://github.com/Maxim-Ivanov/Kindgren_et_al_2019/). Therefore, the pausing signals in our analysis could only have come from the nascent RNA, not from splicing intermediates. When we replotted the plaNET-seq data¹⁷, we performed the smoothing by 10bp to generate the continuous lines for presenting the

Pol II pausing signatures across the sites, as shown in **Supplementary Fig. 4b, c** (now **Supplementary Fig. 6 b, c**).

In Fig6a and 6b, it is striking that promoter-promoter interacting genes were exclusively co-upregulated or co-downregulated during cold treatment. I am skeptical that not all promoter-promoter interacting gene pairs are present in the plots, instead, it is clear that a filter has been applied so that only a subset of gene pairs are shown. Besides, perhaps “22C or 3-hour cold treatment” (line 701) should be “3-hour vs. 22C”, and “3-hour or 12-hour” (line 703) should be “3-hour vs. 12-hour”. Further analysis of these genes “showed that for many gene pairs where one gene was involved in stress response, its co-regulatory gene partner was mainly involved in either cellular metabolic processes or developmental processes” (line279 to 281). Generally, stress responses are considered antagonistic to plant growth and development. The authors should discuss the potential reasons behind this observation.

Response: We thank the Reviewer for this suggestion. We first determined all the mutual PPIs among all three conditions as primed PPIs. Then we determined the differentially expressed genes under three conditions. Among all differentially expressed genes, we found that 70% of the significantly upregulated genes and 68% of significantly downregulated genes contained primed PPIs. Then we correlated their differential expression fold changes for each pair of genes with the primed PPIs and found that they were significantly correlated (up to 0.74, Fig. 6a, b). We have also changed “22°C or 3-hour cold treatment” to be “3-hour vs. 22°C”, and “3-hour or 12-hour” to “3-hour vs. 12-hour” in our revised manuscript highlighted in yellow.

The principle trade-off in plants underscores a strategic allocation of resources, directing plants towards either a stress response or growth contingent upon prevailing conditions. The trade-off mechanism exemplifies how resource allocation is dynamically modulated to enhance plant adaptability and fitness, even amid challenging environmental circumstances^{19,20}. The primed PPIs captured in our CAP-C library have revealed a diverse array of pathways, particularly noteworthy of which are pathways associated with stress responses, anatomical structure development, biosynthesis, and metabolite pathways. Within these categories, we identified a delicate interplay of genes that function as positive and negative regulators within their respective pathways. The co-regulation of these genes could possibly be a mechanism for the trade-off phenomenon.

One example lies in the gene pair of *SLY1* and *TLPI*. *SLY1* promotes growth by positively regulating gibberellin signalling²¹, while *TLPI* acts as a negative regulator of ABA signalling and stress responses²². This strategic partnership embodies the trade-off principle. The co-regulations of both genes show how resources are strategically orchestrated to achieve a balance between growth and stress defence. Another example is the interplay between *FAF1* and *RTP7*. *FAF1*, acting as a negative regulator of plant meristem growth²³, forms a PPI with *RTP7*, which functions in enhancing stress and pathogen responses²⁴. This PPI interaction highlights the balance between allocating resources for growth and prioritizing effective stress adaptation. We have added our reasoning to support our observation regarding this trade-off co-regulations in the discussion section of our revised manuscript in lines 396-404, highlighted in yellow.

Reviewer #2 (Remarks to the Author):

I don't understand why this manuscript could be considered for reviewed in a nature journal. The Cap-C method has already been published in nature biotechnology. This is just applying it to plant? Is that acceptable now?

Response: While we thank the Reviewer for spending time and effort in providing us with many useful comments, we do not agree with these comments.

Firstly, plants have their unique chromatin architectures^{25, 26}. For instance, plants do not contain CTCF- and other insulator-like proteins^{25, 26}. Secondly, plants, as sessile organisms, have evolved with a diverse array of gene regulatory mechanisms to adapt to a vast range of varying environmental conditions²⁷. For example, cold stress is one of the major environmental challenges that affects plant growth. In our study, we demonstrated the dynamic changes of local chromatin contacts in plant response to cold. Therefore, plants are ideal systems to study the dynamics of chromatin architectures in response to diverse environmental conditions, which are important for the currently changing climate. Thirdly, we have discovered the chromatin interactions within the gene body that are significantly associated with Pol II activities across initiation, pausing and termination sites. We found that these associations changed dynamically under different temperature conditions. Fourthly, we have also discovered that plants adopt promoter-promoter co-regulatory networks to enhance adaptability and fitness within diverse environmental contexts.

These exciting discoveries are unique to chromatin features in plants that may be critical to facilitating their adaptation across diversely challenging environmental conditions.

I also don't understand why the author would choose Arabidopsis, which has a small genome and no meaningful long-range chromatin interactions. Yes, Cap-C has very good resolution, which is very useful for large animal genome, and this has been nicely demonstrated in the first nature biotechnology paper. But plants, Arabidopsis in particular do not have such chromatin loops. That's why the author couldn't find anything meaningful interaction from their Cap-C at this super high resolution. They then turned to making a "promoter-promoter-interaction" (PIP) network and "transcription-initiation-cluster" (TIC). But those are just highly expressed genes in the transcription factories that got crosslinked together. These "interaction" has no regulatory function. Why would anyone bother to examine those?

Response: Again, respectfully we do not agree with this particular comment from the Reviewer, and here is our justification.

Firstly, we thank the Reviewer for acknowledging CAP-C is a high resolution of the CAP-C method. In general, high-resolution methods such as our CAP-C⁹, BL-Hi-C¹⁰ and micro-C¹¹ tend to capture more short-distance chromatin interactions as compared to long-distance chromatin interactions. In the original CAP-C paper, the authors harnessed this advantage by applying the method to *Drosophila*, a small-genome organism, and they successfully captured the chromatin contacts that are important for the *Drosophila* development. Indeed, *Arabidopsis* is another ideal organism for applying the CAP-C method due to its compact genome. The *Arabidopsis* genome purportedly used local short-distance chromatin contacts to facilitate its chromatin architecture in the absence of long-distance chromatin contacts, such as TAD domains²⁵. Using CAP-C, we discovered a novel gene regulatory level in *Arabidopsis* chromatin contacts, posing exciting new insights into its genome.

Among all PPIs identified in our study, we plotted the distribution of PPI gene densities over their gene expression levels under three different conditions where we found relatively even distributions (**Rebuttal Figure. 12a**). Highly expressed genes do not have more PPIs (**Rebuttal Figure. 12a**). To further confirm it, we binned the genes based on their strength of their PPIs and plotted their corresponding average gene expression levels, where we found similar, relatively

even distributions without the preference of highly expressed genes (ANOVA, p values = 0.16, 0.12, 0.16) (**Rebuttal Figure. 12b-d**). Our results provide evidence that our CAP-C is able to capture the chromatin contacts independent from gene expression levels. Thus, we did not specifically classify transcription-initiation clusters (TICs) in our study. Additionally, we identified extensive PPIs that existed mutually under three different conditions. These PPIs were not reduced or disappeared when gene expressions were significantly downregulated under cold treatments (**Figure 6**). Therefore, we conclude that these PPIs are likely to be primed across the genome to facilitate the co-regulation of gene expression.

Rebuttal Figure 12. The relationship between PPIs and gene expression levels.

a The line plot showed the distributions of PPI gene densities over their gene expression levels under three different conditions. Red line for the 22°C treatment, blue line for the 4°C treatment

for 3 hours, and green line for the 4°C treatment for 12 hours. The x-axis represents the gene expression level (Transcript per million, TPM), and the y-axis represents the PPI density normalized by the total PPI density. **b-d** The box plots showing the average gene expression levels over different strengths of PPIs under three different conditions. The genes are sorted in ascending order based on the strengths of PPIs and grouped into 1000 bins with 25 genes each. The corresponding average gene expression levels were profiled accordingly. The x-axis represents the bin number. The y-axis represents the gene expression level (Transcript per million, TPM) (ANOVA, p values = 0.16, 0.12, 0.16).

The last sentence of the abstract said "...contributed to transcriptional reprogramming, enhancing our understanding of chromatin conformation-associated gene regulation". None of the result in this paper can demonstrate that such chromatin interaction has any regulatory function.

There are too many papers describing this type of chromatin interaction in plants. But no one can confidently say that that interaction has any regulatory function. Very few exceptional cases in some large plant genomes, in which a few real enhancers can be positioned hundreds of kb away from the target gene, like b1 and tb1. Those rare cases have been extensively studied and people have published those many years ago. Now, if anyone wants to claim that they find new long-range chromatin interaction that can control gene expression, they can't just show there is an interaction by sequencing. because adjacent accessible chromatin or transcribed genes can be cross-linked in such experiment and detected as if they are interacting. if we want to make such functional claim, we must produce evidence. So far, no one is able to do so. lots of people tried to use CRISPR to remove those so-call distal enhancers" and found no meaningful effect on gene expression. That's why we won't consider the plant chromatin interaction would have the same function as those in the animal nucleus.

Response: Again, while we thank the Reviewer for spending time and effort in reviewing our manuscript, respectfully we do not agree with this comment. Here is our justification.

It's widely recognized that promoter-promoter (P-P) and enhancer-promoter (E-P) interactions play a pivotal role in gene regulation. Apart from the tb1 and b1 elements mentioned by the Reviewer, a plethora of other regulatory elements have been substantiated across yeast, animals, as well as in plants^{28, 29, 30, 31, 32, 33, 34, 35, 36, 37, 38, 39, 40, 41, 42, 43, 44, 45, 46}. Examples include the enhancers

associated with the *Pea plastocyanin (PetE)* gene in peas^{36, 42}, the interaction between the *UNBRANCHED3* gene with its Distal Enhancer, the KRN4 element in maize⁴³, NIN^{CE} for NIN expression in *Medicago*⁴⁵, distal CCAAT/NUCLEAR FACTOR Y^{29, 30, 33} for *FLOWERING LOCUS T (FT)*, the Region C for the *LATERAL SUPPRESSOR(LAS)*³² in *Arabidopsis*, and the promoter-promoter interactions between ME2 and MYC2 in tomatoes⁴⁴. It is anticipated that more individual validations will continue to emerge in support of the regulatory roles of promoter-promoter (P-P) and enhancer-promoter (E-P) interactions in plants.

To further validate the regulatory roles of our primed PPIs, we used CRISPR in the protoplast transformation^{47, 48} to delete the regions responsible for our identified PPIs in the gene pairs illustrated in **Figure 6d**. Among three gene pairs, we successfully deleted the regions responsible for our identified PPIs in two sets of these gene pairs. For the gene pair *AT1G80940* and *AT1G80950* connected by PPIs, guide RNAs were designed to target and delete a 590-bp region within the promoter of *AT1G80940* across the PPI sites (**Rebuttal Figure 13a**). This deletion was confirmed by gel analysis and Sanger sequencing (**Rebuttal Figure 13b, c**). In the comparison of 3-hour 4°C vs. 22°C, a distinct co-downregulation trend was observed in the gene pair *AT1G80940* and *AT1G80950* within the wild-type cells (**Rebuttal Figure 13d**). Similarly, in the comparison of 12-hour 4°C vs. 3-hour 4°C, a co-upregulation pattern was observed. In the context of the promoter deletion mutant cells for *AT1G80940*, the gene expression was significantly reduced at 22°C along with significantly reduced cold responses (**Rebuttal Figure 13d**), while the expression of *AT1G80950* was not obviously decreased at 22°C (**Rebuttal Figure 13d**). The expression patterns in plant response to the different cold treatments were very different from those in the wild-type cells (**Rebuttal Figure 13d**). Similarly, we have also validated our identified PPIs in the gene pair of *AT1G78210* and *AT1G78230* (**Rebuttal Figure 13e**). For the gene pair *AT1G78210* and *AT1G78230* connected by PPIs, guide RNAs were designed to target and delete a 566-bp region within the promoter of *AT1G78210* across the PPI sites (**Rebuttal Figure 13f, g**). Co-upregulation of the gene pair *AT1G78210* and *AT1G78230* was evident in the wild-type cells in the comparison of 3-hour 4°C vs. 22°C. A strong down-upregulation of this gene pair could also be seen in wild-type cells in the comparison of 12-hour 4°C vs. 3-hour 4°C (**Rebuttal Figure 13h**). In the context of the promoter deletion mutant cells for *AT1G78210*, the gene expression was significantly reduced at 22°C along with a significantly reduced cold response (**Rebuttal Figure 13h**). Interestingly, in the context of the promoter deletion mutant cells for

AT1G78210, the expression of the PPI paired gene *AT1G78230* was also significantly decreased at 22°C (**Rebuttal Figure 13h**). The expression patterns in plant response to the different cold treatments were dramatically different from those in the wild-type cells (**Rebuttal Figure 13h**). Given that CRISPR deletions exhibit partial effectiveness in protoplast transformation, the significant impacts on the co-regulatory gene expression we observed further underscore the importance of PPIs in co-regulating gene expressions. Taken together, these results indicated that PPIs are important for the gene expressions of these gene pairs and their co-regulations in response to cold.

Rebuttal Figure 13. The disruption of promoter-promoter interactions (PPIs) interferes with the co-regulations of gene expressions.

a and **e** Schematic diagram of CRISPR-Cas9 for *AT1G80940* and *AT1G78210*, respectively. **b** and **f** PCR analysis confirms the deletion. **c** and **g**, DNA sequencing results of the edited band. **d** and **h** qRT-PCR analysis shows gene expression level changes under three conditions. The protoplasts of the control *Col-0* wild type and CRISPR-edited mutant cells were treated at 22°C and 4°C for 3 and 12 hours, respectively. Error bars indicate SEM, asterisks indicate statistically significant differences using *t*-test; * indicates $p < 0.05$, *** indicates $p < 0.001$.

Based on our preliminary results from CRISPR deletions, we further assessed the T-DNA inserted mutants where a previous study suggested that T-DNA insertion is capable of disrupting chromatin interactions²⁸. We identified a T-DNA insertion mutant, the *lpeat1-1* mutant (SALK_065009), with the T-DNA inserted into the promoter of gene *AT1G80950* across the PPI sites (**Rebuttal Figure 14a**). We then confirmed the PPIs detected in our CAP-C in the wild-type plants using 3C and reciprocal 3C experiments^{4, 49} (**Rebuttal Figure 14b, c**). The PPIs between the gene pair *AT1G80940* and *AT1G80950* were disrupted in the T-DNA insertion mutant (**Rebuttal Figure 14b, c**). Similarly, we confirmed another two T-DNA inserted mutants: the *oadlc-1* (SALK_093069) with the T-DNA inserted into the promoter of gene *AT1G78230* across the PPI sites (**Rebuttal Figure 14d**) and the *rssp-1* (SALK_093069) with the T-DNA inserted into the promoter of gene *AT1G60230* across the PPI sites (**Rebuttal Figure 14g**). The identified PPIs between the gene pair *AT1G78210* and *AT1G78230*, and PPIs between the gene pair *AT1G60220* and *AT1G60230* were also disrupted in these T-DNA insertion mutants (**Rebuttal Figure 14e, f, Rebuttal Figure 14h, i**).

Following the validations of abolished PPIs in these T-DNA mutants, we further measured the co-regulation of gene expressions. In the comparison of 3-hour 4°C vs. 22°C in the Col-0 wild-type plants, a distinct co-downregulation trend was observed in the gene pair *AT1G80940* and *AT1G80950*, consistent with the results from the plaNET-seq data in Figure 6 (**Rebuttal Figure 15a, b**). Similarly, in the comparison of 12-hour 4°C vs. 3-hour 4°C, a co-upregulation pattern was observed. In the *lpeat1-1* mutant, the gene expression of *AT1G80950* was significantly reduced at 22°C along with significantly different cold responses (**Rebuttal Figure 15b**). Interestingly, in the *lpeat1-1* mutant, the expression of the PPI paired gene *AT1G80940* was also significantly decreased at 22°C (**Rebuttal Figure 15b**). The plant response expression patterns under different cold treatments were very different from those observed in the wild-type plants (**Rebuttal Figure 15b**). Our results indicated that PPIs are important for gene expressions of the gene pairs and their co-regulations in response to cold treatments. We have also validated our identified PPIs in the gene pair, *AT1G78210* and *AT1G78230* (**Rebuttal Figure 15c, d**). The co-upregulation of the gene pair *AT1G78210* and *AT1G78230* was evident in the wild-type in the comparison of 3-hour 4°C vs. 22°C. A strong co-downregulation of this gene pair could also be seen in the Col-0 wild-type plants in the comparison of 12-hour 4°C vs. 3-hour 4°C (**Rebuttal Figure 15d**). Interestingly, in the *oadlc-1* mutant, the gene expression of *AT1G78230* was not significantly changed at 22°C.

Similarly, in the *oadlc-1* mutant, the expression of the PPI paired gene *AT1G78210* was also not significantly changed at 22°C. Strikingly, the co-regulations of gene expression between the gene pair *AT1G78210* and *AT1G78230* in response to different cold treatments were completely lost (**Rebuttal Figure 15d**). The same validations were also carried out for the gene pair of *AT1G60220* and *AT1G60230* (**Rebuttal Figure 15 e,f**). The co-downregulation of the gene pair *AT1G60220* and *AT1G60230* was evident in the wild-type in the comparison of 3-hour 4°C vs. 22°C. A strong co-upregulation of this gene pair could also be seen in the wild-type plants in the comparison of 12-hour 4°C vs. 3-hour 4°C (**Rebuttal Figure 15 g**). In the *rssp-1* mutant, the gene expression of *AT1G60230* was not significantly changed at 22°C, the same for the PPI paired gene, *AT1G60220*. Strikingly, the co-regulations of gene expression between the gene pair *AT1G60220* and *AT1G60230* in response to different cold treatments were also completely lost. These results further emphasize our conclusion that the co-regulations of gene expressions in plant response to cold were disrupted once the primed PPIs were abolished. Taken together, our new experimental evidence from both CRISPR editing and T-DNA mutants strongly supports the function of the primed PPIs demonstrated from our CAP-C libraries, further strengthening the importance of PPI-mediated co-regulations of gene pair expression.

Rebuttal Figure 14 (New Supplementary Figure 11). 3C experiment detection for PPIs in Col-0 and T-DNA insertion mutants.

a, d, g Schematic diagram of gene pairs of *AT1G80940* and *AT1G80950*. The red arc shows the location of the PPI interaction. The triangle indicates the location of T-DNA insertions within the promoter region: *lpeat1-1* (SALK_065009) for *AT1G80950* gene (*LPEAT1*); *oadlc1-1* (SALK_093069) for *AT1G78230* gene (*Outer arm dynein light chain 1*); *rssp1-1* (SALK_093069)

for *AT1G60230* gene (*Radical SAM superfamily Protein*). The black arrow shows the location of the anchor primer, while black boxes show the locations of the forward primers used in **b**, **e** and **h**, the red arrow shows the primers used in **c**, **f** and **i**.

b, **e**, **h** 3C experiment showing the relative interaction frequency of each fragment with the anchor region, the blue line showing wild-type and the orange line showing the T-DNA insertion mutant.

c, **f**, **i** Reciprocal 3C experiment showing the interaction frequency of the confirmed region from **b** with the reverse primers (shown in red arrows), the blue line showing wild-type and the orange line showing the T-DNA insertion mutant. Errors bars indicate SEM; asterisks indicate statistically significant differences using *t*-test; *** indicates $p < 0.001$.

Rebuttal Figure 15 (New Supplementary Figure 12). Gene expression detection of gene pairs connected by promoter-promoter interactions in response to cold in wild-type and T-DNA insertion mutants.

a Schematic diagram of gene pairs of *AT1G80940* and *AT1G80950*. The red arc shows the location of the PPI interaction. The triangle indicates the location of T-DNA insertion. **b** qRT-PCR analysis of *AT1G80940* and *AT1G80950* under three conditions. **c** Schematic diagram of gene pairs of *AT1G78210* and *AT1G78230*, with the red arc showing the location of the PPI interaction. The triangle indicates the location of T-DNA. **d** qRT-PCR analysis for gene *AT1G78210* and *AT1G78230* under three conditions. **e** Schematic diagram of gene pairs of *AT1G60220* and *AT1G60230*, with the red arc showing the location of the PPI interaction. The triangle indicates the location of T-DNA insertion within the promoter. **f** qRT-PCR analysis for gene *AT1G60220* and *AT1G60230* under three conditions. Errors bars indicate SEM; asterisks indicate statistically significant differences using *t*-test; n.s. not statistically significant; * indicates $p < 0.05$, *** indicates $p < 0.001$.

Reference

1. Kolb HC, Finn MG, Sharpless KB. Click Chemistry: Diverse Chemical Function from a Few Good Reactions. *Angew Chem Int Ed Engl* **40**, 2004-2021 (2001).
2. Liu C, Wang C, Wang G, Becker C, Zaidem M, Weigel D. Genome-wide analysis of chromatin packing in Arabidopsis thaliana at single-gene resolution. *Genome Research* **26**, 1057-1068 (2016).
3. Ariel F, *et al.* R-Loop Mediated trans Action of the APOLO Long Noncoding RNA. *MOLECULAR CELL* **77**, 1055-+ (2020).
4. Crevillén P, Sonmez C, Wu Z, Dean C. A gene loop containing the floral repressor FLC is disrupted in the early phase of vernalization. *Embo j* **32**, 140-148 (2013).
5. Kaul A, Bhattacharyya S, Ay F. Identifying statistically significant chromatin contacts from Hi-C data with FitHiC2. *Nature Protocols* **15**, 991-1012 (2020).
6. Ay F, Bailey TL, Noble WS. Statistical confidence estimation for Hi-C data reveals regulatory chromatin contacts. *Genome Res* **24**, 999-1011 (2014).
7. Yang T, *et al.* HiCRep: assessing the reproducibility of Hi-C data using a stratum-adjusted correlation coefficient. *Genome Research* **27**, 1939-1949 (2017).
8. Robinson JT, *et al.* Integrative genomics viewer. *Nat Biotechnol* **29**, 24-26 (2011).
9. You Q, *et al.* Direct DNA crosslinking with CAP-C uncovers transcription-dependent chromatin organization at high resolution. *Nature Biotechnology* **39**, 225-235 (2021).
10. Li L, *et al.* Global profiling of RNA–chromatin interactions reveals co-regulatory gene expression networks in Arabidopsis. *Nature Plants* **7**, 1364-1378 (2021).
11. Hsieh T-HS, *et al.* Resolving the 3D Landscape of Transcription-Linked Mammalian Chromatin Folding. *Molecular Cell* **78**, 539-553.e538 (2020).
12. Stadhouders R, *et al.* Transcription regulation by distal enhancers: who's in the loop? *Transcription* **3**, 181-186 (2012).
13. Xie Y, *et al.* Enhancer transcription detected in the nascent transcriptomic landscape of bread wheat. *Genome Biology* **23**, 109 (2022).
14. Li G, *et al.* Extensive Promoter-Centered Chromatin Interactions Provide a Topological Basis for Transcription Regulation. *Cell* **148**, 84-98 (2012).
15. Bastian M, Heymann S, Jacomy M. Gephi: An Open Source Software for Exploring and Manipulating Networks. *Proceedings of the International AAAI Conference on Web and Social Media* **3**, 361-362 (2009).
16. Zhu J, Liu M, Liu X, Dong Z. RNA polymerase II activity revealed by GRO-seq and pNET-seq in Arabidopsis. *Nature Plants* **4**, 1112-1123 (2018).
17. Kindgren P, Ivanov M, Marquardt S. Native elongation transcript sequencing reveals temperature dependent dynamics of nascent RNAPII transcription in Arabidopsis. *Nucleic Acids Research* **48**, 2332-2347 (2019).
18. Nojima T, *et al.* Mammalian NET-Seq Reveals Genome-wide Nascent Transcription Coupled to RNA Processing. *Cell* **161**, 526-540 (2015).
19. Monson RK, Trowbridge AM, Lindroth RL, Lerda MT. Coordinated resource allocation to plant growth-defense tradeoffs. *New Phytol* **233**, 1051-1066 (2022).
20. Huot B, Yao J, Montgomery BL, He SY. Growth-defense tradeoffs in plants: a balancing act to optimize fitness. *Mol Plant* **7**, 1267-1287 (2014).
21. Dill A, Thomas SG, Hu J, Steber CM, Sun TP. The Arabidopsis F-box protein SLEEPY1 targets gibberellin signaling repressors for gibberellin-induced degradation. *Plant Cell* **16**, 1392-1405 (2004).

22. Li S, *et al.* A tubby-like protein CsTLP8 acts in the ABA signaling pathway and negatively regulates osmotic stresses tolerance during seed germination. *BMC Plant Biol* **21**, 340 (2021).
23. Wahl V, Brand LH, Guo YL, Schmid M. The FANTASTIC FOUR proteins influence shoot meristem size in *Arabidopsis thaliana*. *BMC Plant Biol* **10**, 285 (2010).
24. Yang Y, *et al.* A mitochondrial RNA processing protein mediates plant immunity to a broad spectrum of pathogens by modulating the mitochondrial oxidative burst. *Plant Cell* **34**, 2343-2363 (2022).
25. Wang C, *et al.* Genome-wide analysis of local chromatin packing in *Arabidopsis thaliana*. *Genome Res* **25**, 246-256 (2015).
26. Rodriguez-Granados NY, *et al.* Put your 3D glasses on: plant chromatin is on show. *JOURNAL OF EXPERIMENTAL BOTANY* **67**, 3205-3221 (2016).
27. López-Maury L, Marguerat S, Bähler J. Tuning gene expression to changing environments: from rapid responses to evolutionary adaptation. *Nature Reviews Genetics* **9**, 583-593 (2008).
28. Zhu B, Zhang W, Zhang T, Liu B, Jiang J. Genome-Wide Prediction and Validation of Intergenic Enhancers in *Arabidopsis* Using Open Chromatin Signatures. *Plant Cell* **27**, 2415-2426 (2015).
29. Liu L, *et al.* Induced and natural variation of promoter length modulates the photoperiodic response of FLOWERING LOCUS T. *Nature Communications* **5**, 4558 (2014).
30. Cao S, Kumimoto RW, Gnesutta N, Calogero AM, Mantovani R, Holt BF, III. A Distal CCAAT/NUCLEAR FACTOR Y Complex Promotes Chromatin Looping at the FLOWERING LOCUS T Promoter and Regulates the Timing of Flowering in *Arabidopsis*. *The Plant Cell* **26**, 1009-1017 (2014).
31. Zhang W, Zhang T, Wu Y, Jiang J. Genome-Wide Identification of Regulatory DNA Elements and Protein-Binding Footprints Using Signatures of Open Chromatin in *Arabidopsis*. *The Plant Cell* **24**, 2719-2731 (2012).
32. Raatz B, *et al.* Specific expression of LATERAL SUPPRESSOR is controlled by an evolutionarily conserved 3' enhancer. *The Plant Journal* **68**, 400-412 (2011).
33. Adrian J, Farrona S, Reimer JJ, Albani MC, Coupland G, Turck F. cis-Regulatory Elements and Chromatin State Coordinately Control Temporal and Spatial Expression of FLOWERING LOCUS T in *Arabidopsis*. *The Plant Cell* **22**, 1425-1440 (2010).
34. McGarry RC, Ayre BG. A DNA element between At4g28630 and At4g28640 confers companion-cell specific expression following the sink-to-source transition in mature minor vein phloem. *Planta* **228**, 839-849 (2008).
35. Yang W, Jefferson RA, Huttner E, Moore JM, Gagliano WB, Grossniklaus U. An egg apparatus-specific enhancer of *Arabidopsis*, identified by enhancer detection. *Plant physiology* **139**, 1421-1432 (2005).
36. Chua YL, Watson LA, Gray JC. The transcriptional enhancer of the pea plastocyanin gene associates with the nuclear matrix and regulates gene expression through histone acetylation. *The Plant Cell* **15**, 1468-1479 (2003).
37. Sidorenko L, Li X, Tagliani L, Bowen B, Peterson T. Characterization of the regulatory elements of the maize P-rr gene by transient expression assays. *Plant molecular biology* **39**, 11-19 (1999).
38. Lund G, Prem Das O, Messing J. Tissue-specific DNase I-sensitive sites of the maize P gene and their changes upon epimutation. *The Plant Journal* **7**, 797-807 (1995).

39. IM van der Meer IM, Spelt CE, Mol JN, Stuitje AR. Promoter analysis of the chalcone synthase (chs A) gene of *Petunia hybrida*: a 67 bp promoter region directs flower-specific expression. *Plant molecular biology* **15**, 95-109 (1990).
40. Nagy F, Boutry M, Hsu M, Wong M, Chua N. The 5'-proximal region of the wheat Cab-1 gene contains a 268-bp enhancer-like sequence for phytochrome response. *The EMBO journal* **6**, 2537-2542 (1987).
41. Timko MP, *et al.* Light regulation of plant gene expression by an upstream enhancer-like element. *Nature* **318**, 579-582 (1985).
42. Simpson J, Timko MP, Cashmore AR, Schell J, Van Montagu M, Herrera-Estrella L. Light-inducible and tissue-specific expression of a chimaeric gene under control of the 5'-flanking sequence of a pea chlorophyll a/b-binding protein gene. *The EMBO Journal* **4**, 2723-2729 (1985).
43. Du Y, *et al.* UNBRANCHED3 Expression and Inflorescence Development is Mediated by UNBRANCHED2 and the Distal Enhancer, KRN4, in Maize. *PLOS Genetics* **16**, e1008764 (2020).
44. Wang H, *et al.* MED25 connects enhancer–promoter looping and MYC2-dependent activation of jasmonate signalling. *Nature Plants* **5**, 616-625 (2019).
45. Liu J, *et al.* A Remote cis-Regulatory Region Is Required for NIN Expression in the Pericycle to Initiate Nodule Primordium Formation in *Medicago truncatula*. *The Plant Cell* **31**, 68-83 (2019).
46. Huang Y, *et al.* HSFA1a modulates plant heat stress responses and alters the 3D chromatin organization of enhancer-promoter interactions. *Nature Communications* **14**, 469 (2023).
47. Sheen J. Ca²⁺-dependent protein kinases and stress signal transduction in plants. *Science* **274**, 1900-1902 (1996).
48. Duque P, Szakonyi D. *Environmental responses in plants: Methods and protocols*. Springer (2016).
49. Jamge S, Stam M, Angenent GC, Immink RGH. A cautionary note on the use of chromosome conformation capture in plants. *Plant Methods* **13**, 101 (2017).

Reviewer #1 (Remarks to the Author):

For the plots in Fig6a and b, my concerns still remain. In fact, the authors' rebuttal letter confirms that the correlation was calculated based on a selection strategy, in which only up- and down-regulated genes forming PPIs were selected. With such a selection strategy, of course, there will be a strong positive correlation because all the data points are restricted to the first and the third quadrants. Therefore, the conclusion (line 289, in comparison to the plot shown in FigS10) doesn't make sense. Instead, the author should revise the analysis pipeline, for instance, to select ALL the PPI-paired genes, in which at least one gene in each pair is considered up-regulated, then, to check fold changes of the other gene. As a control, dummy PPIs can be randomly generated, followed by the same analysis.

At the moment, I feel that Fig6 is misleading. Among a large number of PPIs, one can always spot some gene pairs with concordant expression changes. Whether such coordinated expression change is over-represented in PPI-gene pairs and whether PPI occurs independently from gene expression control is still in question.

The "tandem orientation" (line 150) perhaps refers to PPI, not E-P contacts. (It is hard to determine the "orientation" of a putative enhancer.).

Reviewer #2 (Remarks to the Author):

In the revised manuscript, the authors employed CRISPR/Cas9 to delete a region took part in "promoter-promoter" interaction located upstream of the AT1G80940 gene in Arabidopsis protoplasts (rebuttal fig 13). RT-qPCR showed a down-regulation in the expression of AT1G80940, as well as a concurrent down-regulation of the nearby AT1G80950 gene, whose promoter interacts with AT1G80940. They also conducted a similar experiment on another pair of genes. The authors aim to utilize these findings as supporting evidence for their assertion that promoter-promoter interactions can play a role in gene regulation.

My concern is that only a fraction of the protoplasts would carry this deletion. The gel image provided in Rebuttal Fig 13b and 13f also suggests that only a very small proportion of the protoplasts have undergone editing (comparing the faint edited band at the bottom with the strong unedited band at the top). If this is the case, it implies that the down-regulation of this gene would only occur in a minor percentage of cells. Therefore, I find it puzzling how the RT-qPCR results indicate a significant decrease in gene expression from 1-1.5 to less than 0.2 (rFig 13d and h). This raises doubts about the reliability and reproducibility of the findings. Alternatively, is it possible that something has transpired within the unmodified cells, such as Cas9 binding to the site, hindering other transcription factors from accessing the open chromatin of the promoter? In any case, it appears necessary for the authors to establish a stable transgenic line for the CRISPR/Cas9 experiment and conduct proper RNA-seq analysis to obtain more robust results.

Also, one can not talk about chromatin interaction without considering open/accessible chromatin (ATAC-seq). I suggest the author to read this Arabidopsis Hi-C paper and think about how to present their finding in a sensible manner (<https://doi.org/10.1093/nar/gkad710>).

We thank the reviewers for their time and effort in helping us improve our manuscript. We have fully addressed the reviewers' comments by adding additional new experiments and analyses to further strengthen our conclusions. Please find the detailed responses as follows.

REVIEWER COMMENTS

Reviewer #1 (Remarks to the Author):

For the plots in Fig6a and b, my concerns still remain. In fact, the authors' rebuttal letter confirms that the correlation was calculated based on a selection strategy, in which only up- and down-regulated genes forming PPIs were selected. With such a selection strategy, of course, there will be a strong positive correlation because all the data points are restricted to the first and the third quadrants. Therefore, the conclusion (line 289, in comparison to the plot shown in FigS10) doesn't make sense. Instead, the author should revise the analysis pipeline, for instance, to select ALL the PPI-paired genes, in which at least one gene in each pair is considered up-regulated, then, to check fold changes of the other gene. As a control, dummy PPIs can be randomly generated, followed by the same analysis. At the moment, I feel that Fig6 is misleading. Among a large number of PPIs, one can always spot some gene pairs with concordant expression changes. Whether such coordinated expression change is over-represented in PPI-gene pairs and whether PPI occurs independently from gene expression control is still in question.

Response: Response: We thank the Reviewer for the comments. We apologize for the figure presentation that might lead to the confusion.

The large number of PPIs identified in our CAP-C may have diverse functional roles in gene regulations such as co-regulations of gene expression¹, RNA-mediated chromatin regulations^{2,3}, and chromatin interaction-mediated phase separation⁴. Here, we revealed one of the functional roles as facilitating the co-regulation of gene expression in *Arabidopsis* using our CAP-C. To avoid the confusion, we performed the co-expression analysis followed Peng et al.⁵ in all three conditions. We calculated the Pearson correlation coefficient for expression of all gene pairs involved in PPIs.

Randomly selected gene pairs with the same number and similar distance distribution of the PPI connected gene pairs were built as control groups. We found that the Pearson correlation coefficients for expression of all gene pairs with PPIs for all three conditions were significantly higher than those from the random gene pairs (**Rebuttal Figure 1, New Figure 6a, New Supplementary Figure 11**), indicating that the gene pairs with PPIs tend to be co-expressed. Thus, our results suggest that the primed PPIs may contribute to the co-regulations of gene expression in response to cold treatments.

We further validate these results using individual gene pairs with PPIs. Our experimental evidence from both T-DNA mutants (**Supplementary Figure 12-13**) and CRISPR stable transgenic mutants (**Rebuttal Figure 3, New Supplementary Figure 14**) have demonstrated the importance of PPI in the co-regulations of gene pair expression. We have revised our manuscript accordingly in line 284-292, highlighted in yellow.

Rebuttal Figure 1. (New Figure 6a, New Supplementary Figure 11) Pearson correlation coefficients of expressions from random gene pairs and PPI connected gene pairs in 22°C(a), cold treatment for 3 hours(b) and 12 hours(c). The Pearson correlation coefficients for gene pairs connected by promoter-promoter interaction (PPI) are significantly higher compared to those for randomly selected genes pairs⁵. Randomly selected gene pairs with the same number and similar distance distribution of the PPI connected gene pairs were built as control groups. Random gene pairs A: random gene pairs with the same number; random gene pairs B: random gene pairs with similar distance distribution of PPI gene pairs. Both random procedures were repeated 1000 times. $p < 0.0001$ from t -test.

The “tandem orientation” (line 150) perhaps refers to PPI, not E-P contacts. (It is hard to determine the “orientation” of a putative enhancer.).

Response: We thank the Reviewer for the comments. We have changed the term and did analysis of enhancer upstream and downstream of the connected promoters in the proximal and distal group separately. Among proximal contacts, 62% were upstream and 38% downstream of the interacting promoter, while distal contacts exhibited 51% up upstream and 49% downstream orientation (**Rebuttal Figure 2, New Supplementary Figure 4**). We have modified the description in the revised manuscript accordingly in line 148-150, highlighted in yellow.

Rebuttal Figure 2 (New Supplementary Figure 4). Proportions of different types of E-P chromatin contacts. The proportion of enhancers upstream and downstream of the connected promoters in both proximal and distal E-P chromatin contacts.

Reviewer #2 (Remarks to the Author):

In the revised manuscript, the authors employed CRISPR/Cas9 to delete a region took part in "promoter-promoter" interaction located upstream of the AT1G80940 gene in Arabidopsis protoplasts (rebuttal fig 13). RT-qPCR showed a down-regulation in the expression of AT1G80940, as well as a concurrent down-regulation of the nearby AT1G80950 gene, whose promoter interacts with AT1G80940. They also conducted a similar experiment on another pair of genes. The authors aim to utilize these findings as supporting evidence for their assertion that promoter-promoter interactions can play a role in gene regulation.

My concern is that only a fraction of the protoplasts would carry this deletion. The gel image

provided in Rebuttal Fig 13b and 13f also suggests that only a very small proportion of the protoplasts have undergone editing (comparing the faint edited band at the bottom with the strong unedited band at the top). If this is the case, it implies that the down-regulation of this gene would only occur in a minor percentage of cells. Therefore, I find it puzzling how the RT-qPCR results indicate a significant decrease in gene expression from 1-1.5 to less than 0.2 (rFig 13d and h). This raises doubts about the reliability and reproducibility of the findings. Alternatively, is it possible that something has transpired within the unmodified cells, such as Cas9 binding to the site, hindering other transcription factors from accessing the open chromatin of the promoter? In any case, it appears necessary for the authors to establish a stable transgenic line for the CRISPR/Cas9 experiment and conduct proper RNA-seq analysis to obtain more robust results.

Response: We thank the Reviewer for the comments. We would emphasize that in our previous revision, we have also characterized the corresponding T-DNA inserted mutants (**Supplementary Figure 12-13**). Previous study has already suggested that T-DNA insertion is capable of disrupting chromatin interactions⁶. We identified three T-DNA insertion mutants where the corresponding PPIs were disrupted in these T-DNA insertion mutants (**Supplementary Figure 12**). The plant response expression patterns under different cold treatments in these T-DNA insertion mutants were very different from those observed in the wild-type plants (**Supplementary Figure 13**). Our results strongly indicated that PPIs are important for gene expressions of the gene pairs and their co-regulations in response to cold treatments.

To further validate the effect of PPI disruption, we also successfully generated the CRISPR/Cas9 stable transgenic lines for the two sets of gene pairs connected by PPIs. For the gene pair *AT1G80940* and *AT1G80950* connected by PPIs, we obtained a CRISPR/Cas9 stable transgenic line with a 512-bp region within the promoter of *AT1G80940* across the PPI sites (**Rebuttal Figure 3, New Supplementary Figure 14**). This deletion was confirmed by gel analysis and Sanger sequencing (**Rebuttal Figure 3b, c**). In the context of the promoter deletion mutants for *AT1G80940*, the expression patterns of both genes in plant response to the different cold treatments were very different from those in the wild-type plants (**Rebuttal Figure 3d**). Similarly, we have also validated our identified PPIs in the gene pair of *AT1G78210* and *AT1G78230* (**Rebuttal Figure 3e**). For the gene pair *AT1G78210* and *AT1G78230* connected by PPIs, guide RNAs were designed to target and delete a 565-bp region within the promoter of *AT1G78210* across the PPI

sites (**Rebuttal Figure 3f,g**). In the promoter deletion mutants for *AT1G78210*, the expression patterns of both genes in response to cold were also dramatically different from those in the wild-type plants (**Rebuttal Figure 3h**). The significant impacts on the co-regulatory gene expression we observed further underscore the importance of PPIs in co-regulating gene expressions. Taken together, these results indicated that PPIs are important for the gene expressions of these gene pairs and their co-regulations in response to cold. We have revised our manuscript accordingly in line 343-360, highlighted in yellow.

Rebuttal Figure 3 (New Supplementary Figure 14) Gene expression detection of genes pairs connected by PPIs in response to cold in wild-type and CRISPR–Cas9-based deletion mutants.

a and **e** Schematic diagram of CRISPR-Cas9-based deletions for *AT1G80940* and *AT1G78210*, respectively. **b** and **f** PCR analysis confirms the deletion. **c** and **g**, DNA sequencing results of the edited band. **d** and **h** qRT-PCR analysis shows gene expression level changes under three conditions. Wild-type plants and CRISPR-Cas9-based deletion mutants were treated at 22°C and 4°C for 3 and 12 hours, respectively. Three independent biological replicates were assessed, error bars indicate SEM, asterisks indicate statistically significant differences using *t*-test; n.s, not significant; * indicates $p < 0.05$, *** indicates $p < 0.001$.

Also, one can not talk about chromatin interaction without considering open/accessible chromatin (ATAC-seq). I suggest the author to read this Arabidopsis Hi-C paper and think about how to present their finding in a sensible manner (<https://doi.org/10.1093/nar/gkad710>).

Response: We thank the Reviewer for the comments. Indeed, we also observed an association between our CAP-C data and ATAC-seq data. In **Rebuttal Figure 4a**, we compared the ATAC-seq signals between the top 10% genes with the highest chromatin interaction levels (high CAP-C signals) and the bottom 10% genes with the lowest chromatin interaction levels (low CAP-C signals). We observed that the top 10% genes have much stronger ATAC-seq signals in comparison with the bottom 10% genes. Similarly, in **Rebuttal Figure 4b**, we compared the CAP-C data between the top 10% genes with the most chromatin accessibility (high ATAC-seq signals) and the bottom 10% genes with the least chromatin accessibility (low ATAC-seq signals). We also found that the top 10% genes have much stronger chromatin interactions in comparison with the bottom 10% genes. Our results indicate the chromatin accessibility may facilitate the local chromatin interactions. This association between chromatin accessibility and chromatin interactions (**Supplementary Figure 5**) may indicate the gene folding between TSS and the TTS⁷.

We have added the part of results into our revised manuscript in line 181-183 and 400-402, highlighted in yellow.

Rebuttal Figure 4 (New Supplementary Figure 5). CAP-C association between chromatin accessibility signals and chromatin contacts at gene locus level. a. Genes were categorized

based on the enrichment of CAP-C data across the gene body. The 10% of genes with the highest level of CAP-C enrichment (represented by orange line) and the lowest level of CAP-C enrichment (represented by green line) were used to plot ATAC-seq signal profiles⁸ within the 3Kb flanking region of the gene body, n=2648. **b.** Meta-profile shows CAP-C interaction signal profiles with high (red) and weak (blue) ATAC-seq signal enrichment, n=2682. Genes were aligned by the transcription start site (TSS) and the transcription termination site (TTS).

References:

- 1 Li, G. *et al.* Extensive promoter-centered chromatin interactions provide a topological basis for transcription regulation. *Cell* **148**, 84-98 (2012). <https://doi.org/10.1016/j.cell.2011.12.014>
- 2 Li, L. *et al.* Global profiling of RNA–chromatin interactions reveals co-regulatory gene expression networks in Arabidopsis. *Nature Plants* **7**, 1364-1378 (2021). <https://doi.org/10.1038/s41477-021-01004-x>
- 3 Li, X. & Fu, X.-D. Chromatin-associated RNAs as facilitators of functional genomic interactions. *Nature Reviews Genetics* **20**, 503-519 (2019). <https://doi.org/10.1038/s41576-019-0135-1>
- 4 Mora, A., Huang, X., Jauhari, S., Jiang, Q. & Li, X. Chromatin Hubs: A biological and computational outlook. *Computational and Structural Biotechnology Journal* **20**, 3796-3813 (2022). <https://doi.org/https://doi.org/10.1016/j.csbj.2022.07.002>
- 5 Peng, Y. *et al.* Chromatin interaction maps reveal genetic regulation for quantitative traits in maize. *Nature Communications* **10**, 2632 (2019). <https://doi.org/10.1038/s41467-019-10602-5>
- 6 Zhu, B., Zhang, W., Zhang, T., Liu, B. & Jiang, J. Genome-Wide Prediction and Validation of Intergenic Enhancers in Arabidopsis Using Open Chromatin Signatures. *Plant Cell* **27**, 2415-2426 (2015). <https://doi.org/10.1105/tpc.15.00537>
- 7 Lee, H. & Seo, Pil J. Accessible gene borders establish a core structural unit for chromatin architecture in Arabidopsis. *Nucleic Acids Research* **51**, 10261-10277 (2023). <https://doi.org/10.1093/nar/gkad710>
- 8 Potok, M. E. *et al.* Arabidopsis SWR1-associated protein methyl-CpG-binding domain 9 is required for histone H2A.Z deposition. *Nature Communications* **10**, 3352 (2019). <https://doi.org/10.1038/s41467-019-11291-w>

Reviewer #1 (Remarks to the Author):

Upon thorough examination, I am pleased to affirm that the authors have attended to the issues I raised and have made appropriate revisions to their manuscript. I have no more concerns.

Reviewer #2 (Remarks to the Author):

i have no further concerns. Just one suggestion, the Sup Fig 14 is very nice. Maybe the author can consider put one of its sup panel, say move panel A and D from Sup Fig14 to their main figure 6?

REVIEWERS' COMMENTS

Reviewer #1 (Remarks to the Author):

Upon thorough examination, I am pleased to affirm that the authors have attended to the issues I raised and have made appropriate revisions to their manuscript. I have no more concerns.

Response:

We are grateful for the Reviewer's confirmation that all concerns have been addressed. We are also very grateful for their suggestions in helping us improve our manuscript.

Reviewer #2 (Remarks to the Author):

i have no further concerns. Just one suggestion, the Sup Fig 14 is very nice. Maybe the author can consider put one of its sup panel, say move panel A and D from Sup Fig14 to their main figure 6?

Response:

We appreciate the reviewer's positive feedback on our response letter, particularly regarding Supplementary Figure 14. Due to the figure size of the main figure 6, we would keep the original arrangement of figures.